



**Spatial gradients in soil-carbon character of a coastal forested floodplain are associated**
**with abiotic features, but not microbial communities**
Aditi Sengupta[1*], Julia Indivero[2], Cailene Gunn[2], Malak M. Tfaily[3,4], Rosalie K. Chu[4], Jason
Toyoda[4], Vanessa L. Bailey[1], Nicholas D. Ward[2,5], James C. Stegen[1]
1. Biological Sciences Division, Pacific Northwest National Laboratory, PO Box 999

7          MSIN: J4-18, Richland, WA 99352

2. Marine Sciences Laboratory, Pacific Northwest National Laboratory, 1529 West Sequim

9          Bay Road, Sequim, WA 98382

3.  Department of Soil, Water, and Environmental Sciences, University of Arizona, Tucson,

11         AZ 85719

4. Environmental Molecular Sciences Laboratory, Pacific Northwest National Laboratory,

13         Richland, WA 99352

5. School of Oceanography, University of Washington, Seattle, WA 98195
Correspondence to: aditi.sengupta@pnnl.gov



**Abstract**
Coastal terrestrial-aquatic interfaces (TAIs) are dynamic zones of biogeochemical cycling
influenced by salinity gradients. However, there is significant heterogeneity in salinity influences
on TAI soil biogeochemical function. This heterogeneity is perhaps related to unrecognized
mechanisms associated with carbon (C) chemistry and microbial communities.  To investigate
this potential, we evaluated hypotheses associated with salinity-associated shifts in organic C
thermodynamics, biochemical transformations, and heteroatom content in a first-order coastal
watershed in the Olympic Peninsula of Washington state, USA. In contrast to our hypotheses,
thermodynamic favorability of water soluble organic compounds in shallow soils decreased with
increasing salinity, as did the number of inferred biochemical transformations and total
heteroatom content. These patterns indicate lower microbial activity at higher salinity that is
potentially constrained by accumulation of less favorable organic C. Furthermore, organic
compounds appeared to be primarily marine/algal-derived in forested floodplain soils with more
lipid-like and protein-like compounds, relative to upland soils that had more lignin-, tannin-, and
carbohydrate-like compounds. Based on a recent simulation-based study, we further
hypothesized a relationship between microbial community assembly processes and C chemistry.
Null modelling revealed strong influences of dispersal limitation over microbial composition,
which may be due to limited hydrologic connectivity within the clay-rich soils. Dispersal
limitation indicated stochastically assembled communities, which was further reflected in the
lack of an association between community assembly processes and C chemistry. This suggests a
disconnect between microbial community composition and C biogeochemistry, thereby
indicating that the salinity-associated gradient in C chemistry was driven by a combination of
spatially-structured inputs and salinity-associated metabolic responses of microbial communities
that were independent of community composition. We propose that impacts of salinity on coastal



soil biogeochemistry need to be understood in the context of C chemistry,
hydrologic/depositional dynamics, and microbial physiology, while microbial composition may
have less influence.
**1.  Introduction**
The interface between terrestrial and aquatic ecosystems represent a dynamic and poorly
understood component of the global carbon (C) cycle, particularly along the tidally-influenced
reaches of coastal watersheds where terrestrial and marine biospheres intersect (Krauss et al.,
2018; Neubauer et al., 2013; Tank et al., 2018; Ward et al., 2017b). Moreover, the nutrient cycles
occurring at these terrestrial-aquatic interfaces (TAIs) influence locally important ecosystem
services like contaminant fate and transport and water quality (Conrads and Darby, 2017; Vidon
et al., 2010). While coastal soil C stocks are being increasingly quantified (Hinson et al., 2017;
Holmquist et al., 2018; Krauss et al., 2018), the impact of tidally-driven salinity gradients on
molecular level features of the soil-C pool and the processes driving soil organic matter (OM)
cycling are poorly studied (Barry et al., 2018; Hoitink et al., 2009; Sawakuchi et al., 2017; Ward
et al., 2017b), particularly in settings with low freshwater inputs that allows significant seawater
intrusion compared to large river systems (Hoitink and Jay, 2016). Moreover, there is some
indication that microbial diversity and composition impact soil C storage and mineralization
(Mau et al., 2015; Trivedi et al., 2016). This points to the intriguing possibility that processes
governing microbial community assembly may be associated with OM chemistry, but
evaluations of such associations are lacking. This lack of mechanistic knowledge combined with
significant ecosystem heterogeneity in biogeochemical function across salinity gradients (more
below), highlights a need to understand how molecular-level processes vary with seawater
exposure along coastal TAIs. Doing so will help enhance predictive models of TAI



biogeochemistry that can be potentially included in ecosystem models to more accurately
represent the role of TAIs in the broader Earth system (U.S. DOE., 2017).

Modeling of coastal TAIs is currently impeded by poor knowledge of the mechanisms
underlying salinity-driven variation in biogeochemical function of associated soils. Previous
studies have evaluated function primarily as carbon dioxide ($CO_2$) and methane ($CH_4$) flux
measurements from soil, and/or soil OM concentrations measured as bulk soil C, percent OM
and porewater dissolved organic C (DOC) concentrations in large scale coastal plain river
systems. Results from field-based natural salinity gradient studies, long-term field manipulations
of salinity exposure, and lab-based incubation studies subjecting soils to varying levels of
salinity broadly show the following trends: increases in $CO_2$ and decreases in $CH_4$ emissions in
freshwater soils exposed to increasing salinity (Chambers et al., 2011, 2013, 2014; Liu et al.,
2017; Marton et al., 2012; Neubauer et al., 2013; Steinmuller and Chambers, 2018; Weston et al.,
2006, 2011), and decreases in $CO_2$ and $CH_4$ emissions from soils with a natural history of being
exposed to saline environment when exposed to elevated salinity (Chambers et al., 2013; Herbert
et al., 2018; Neubauer et al., 2005, 2013; Weston et al., 2014) (also see Table S1). Two
exceptions have been observed where $CO_2$ emissions decreased in historically freshwater coastal
wetland soils exposed to seawater (Ardón et al., 2018; Herbert et al., 2018). These observations
suggest that microbial activity usually increases with salinity in soils that were not previously
exposed to saline conditions, while simultaneously indicating reduced microbial activity with
increasing salinity in soils that have a historical exposure to elevated salinity. In contrast to
relatively consistent responses of gas fluxes to changes in salinity, there are strong
inconsistencies in DOC responses, including no change (Weston et al., 2006, 2011, 2014),



increased DOC (Chambers et al., 2014; Tzortziou et al., 2011), and decreased DOC (Ardón et al.,
2016, 2018; Liu et al., 2017; Yang et al., 2018) with increasing salinity.

Relatively consistent gas flux responses combined with inconsistent DOC responses of soils
exposed to elevated salinity suggest at least a partial decoupling between microbially driven
biogeochemical rates and the concentration of DOC. This apparent decoupling between the size
of the C pool and microbial activity suggests that C biogeochemistry is influenced by salinity-
exposure history, which in turn influences nutrient resources available to soil microbial
communities. Specifically, any systematic shifts in soil organic carbon (SOC) chemistry profiles
that occur along natural salinity gradients (Bischoff et al., 2018; Neubauer et al., 2013), which
cannot be observed with bulk C measurements may result in unpredictable carbon fluxes.
Moreover, bulk C content can show no change across gradients of salinity (Neubauer et al.,
2013) and may fail to capture an integrated view of microbial-activity driven C cycling dynamics
at TAIs. In contrast, detailed molecular-level evaluation of SOC composition can provide a more
mechanistic view of OC transformations, relative to bulk measures of C content or gas flux
measurements. Analyses of specific chemical biomarkers such as lignin phenols, amino acids,
and lipids have been used in soils, sediments, and water to quantitatively evaluate the provenance
of terrestrial-derived OM (Hedges et al., 1997), the reactivity of OM as it travels through a soil
column (Shen et al., 2015), and microbial community composition (Langer and Rinklebe, 2009),
respectively. While biomarkers provide quantitative details on OC cycling, they generally
represent a small fraction of the total OM pool, thus, non-targeted approaches such as analysis of
thousands of peaks via Fourier Transform Ion Cyclotron Resonance Mass Spectrometry (FTICR-
MS) have become increasingly widespread for determining molecular-level organic compound
signatures (Rivas-Ubach et al., 2018) across a variety of terrestrial (Bailey et al., 2017; Simon et





al., 2018) , aquatic/marine (Lechtenfeld et al., 2015), and transitional settings such as hyporheic
zones (Graham et al., 2017a) and river-ocean gradients (Medeiros et al., 2015).

Despite its potential importance, a detailed understanding of the characteristics of soil organic
compounds (Zark and Dittmar, 2018) and their association with microbial communities in
coastal TAIs is currently not available. However, starting with an assumption of increases in
microbial activity with increasing salinity (Nyman and Delaune, 1991; Smith et al., 1983;
Tzortziou et al., 2011) provides a series of expectations. First, it is generally expected that
microbes preferentially degrade compounds with higher nominal oxidation states (NOSC) or
lower Gibbs Free Energy ($\Delta G^0_{Cox}$) due to greater thermodynamic favorability (Boye et al., 2017;
Ward et al., 2017a), although factors such as redox state, mineral associations, and microbial
community composition can alter this generality (Schmidt et al., 2011). The basic assumption
that OM reactivity follows NOSC leads to the expectation that the average $\Delta G^0_{Cox}$ of the
compounds in the resource environment will increase with increasing salinity as organic
compounds with greater thermodynamic favorability are preferentially depleted (LaRowe and
Van Cappellen, 2011). Second, actively growing microbial communities are known to enhance
biochemical transformations and generate heteroatom containing organic molecules [sulfur (S),
nitrogen (N) and phosphorus (P)] (Guillemette et al., 2018; Koch et al., 2014; Ksionzek et al.,
2016); therefore greater heteroatom content and more biochemical transformations are expected
with increasing salinity. Third, studies have also shown that microorganisms adapt to saline
conditions through the production or sequestration of osmolytes (Gouffi et al., 1999; Roberts,
2005; Sleator and Hill, 2002),  a strategy that requires organic N mining. This suggests a
potential increase in N-containing biochemical transformation with increasing salinity. Fourth,
soil OM in saturated environments like floodplains are expected to be less oxygenated and can



also receive deposition of suspended sediments during flooding, both of which may result in a
greater abundance of marine/algal derived OM exhibiting low oxygen to carbon (O/C) and high
hydrogen to carbon (H/C) ratio as compared to upland soils (Seidel et al., 2016; Tfaily et al.,
2014; Ward et al., 2019b). We therefore expect a greater relative abundance of lipids and
proteins and less lignin and tannin compounds in the floodplain soils, relative to upland (i.e.,
drained) soil.

While we expect systematic shifts in C chemistry across landscape scale salinity gradients, an
open question is the degree to which C chemistry is associated with ecological assembly
processes governing microbial communities. Soil microorganisms transform soil C, but there is
limited evidence of direct links between microbial community assembly processes and
molecular-level soil C chemistry (Kubartová et al., 2015; Rocca et al., 2015; Trivedi et al., 2016;
van der Wal et al., 2015).  Assembly processes, broadly divided into deterministic (selective) and
stochastic (random) factors, function over space and time to structure microbial communities,
which in turn mediate biogeochemical cycles (Graham et al., 2016, 2017b; Nemergut et al.,
2013a; Stegen et al., 2015). These processes can be inferred from phylogenetic distances among
microbial taxa using ecological null models, which have been widely employed to understand
community assembly processes in subsurface microbial ecology (Caruso et al., 2011; Dini-
Andreote et al., 2015; Graham et al., 2017a, 2018; Stegen et al., 2012).  Furthermore, a recent
study used ecological simulation modeling to show that environments experiencing increasing
rates of dispersal processes are linked to reduced biogeochemical functioning (Graham and Fine,
2008). This leads to the hypothesis that the influence of deterministic selection will progressively
increase with salinity-driven increases in microbial activity.





The objective of the current study was to test the following hypotheses in a coastal forested
floodplain and adjacent upland forest: (i) the overall Gibbs Free Energy of organic compounds
will increase with increasing salinity; (ii) biochemical transformations, heteroatom content, and
N-containing biochemical transformation will increase with increasing salinity; (iii) the lipid and
protein compound classes will be prevalent in the floodplain soils compared to upland soils in
which lignin- and tannin-type molecules will dominate; and (iv) microbial community assembly
processes will be increasingly deterministic as salinity increases. The chemical forms of C in
these soils were characterized using FTICR-MS. We also employed ecological null model
analysis to evaluate the relationship between C chemistry and the influences of assembly
processes on microbial communities. Based on our results, we propose a conceptual model of
organic C processing in a coastal forested floodplain in which landscape-scale gradients in C
chemistry are driven by a combination of spatially-structured inputs and salinity-associated
metabolic responses of microbial communities that are independent of community composition.

**2.  Materials and Methods**:
**2.1 Site Information and Soil Sampling**
Soils along a coastal watershed draining a small first order stream, Beaver Creek, in the
Washington coast were selected for this study. Beaver Creek is a tributary of Johns River and
experiences a high tidal range of up to 2.5 m that extends midway up the first-order stream's
channel and inundates the landscape in its floodplains. The confluence of Beaver Creek and
Johns River is roughly 2.5 km upstream of the Grays Harbor estuary and 14.5 km from the
Pacific Ocean, and experiences variable exposure to saline waters at high tide (Fig. 1). Surface
water salinity near Beaver Creek's confluence ranges from 0 psu at low tide to 30 psu at high
tide during dry periods (Ward, unpublished). Tidal exchange to Beaver Creek was restored after



2014 when a culvert near the creek's confluence with Johns River was removed (Washington
Department of Fish and Wildlife, 2019). Due to the minimal past tidal exchange, the floodplain
is dominated by gymnosperm trees (*Picea sitchensis*) that are rapidly dying since the culvert
removal (Ward et al., 2019a). The headwaters (before the river channel forms) is a sparsely
forested, perennially inundated freshwater wetland with tidal exchange blocked by a beaver dam,
followed downstream by a densely forested setting along the river channel. Towards Beaver
Creek's confluence salt tolerant grasses such as *Agrostis stolinifera* become the most dominant
land cover as forest cover becomes more sparse. The watershed's hillslope/uplands is dominated
by *Tsuga heterophylla* trees, but *Picea sitchensis* are also present.

Two sampling transects perpendicular to the river along the up/downstream salinity gradient
were established and represent a high salt exposure site close to the culvert breach location and a
moderate salt exposure site upstream of the high salt exposure site. These transects represent a
coastal forested wetland with brackish (semi-salty) groundwater and consisted of three terrestrial
sampling points at each transect extending from the riparian zone to the beginning of the steep
upslope. An additional soil sampling point ~20m uphill from the moderate salt exposure site
transect served as a purely terrestrial upland endmember. The floodplain transects represented
hydric soils classified as Ocosta silty clay loam while the upland site was a well-drained Mopang
silt loam. The transects experience periodic inundation episodes which result in surface pooling
of tidal water.

Soil samples were collected in triplicate at each of the seven locations (Fig. 1) [BC2, BC3, and
BC4 at the high-salt exposure transect, locations BC12, BC13, BC14 at the moderate salt
exposure transect, and BC15 as upland site]. The high-salt exposure transect was 230 m from the





moderately saline transect (0.6 km from the confluence of Beaver Creek with Johns River), and
each site at the transect was ~25 m apart from the next. For data comparison's sake, we classify
BC2, BC3, BC12, and BC13 as **floodplain** sites while BC4 and BC14 are further **inland** and ~75
m away from the creek at the base of the densely wooded hillslope. Soil samples for molecular
characterization studies were collected at two depths—shallow (10 cm) and deep (19-30 cm).
Samples were collected from the face of soil pits using custom mini-corers, placed into sterile
amber glass vials, purged with $N_2$ to maintain anaerobic conditions, frozen in the field within an
hour at -20 °C, and stored at -80 °C on return to the lab. Bulk samples were collected for soil
physicochemical characterization including texture classification with hydrometer method after
organic matter removal, dry combustion with direct measure of total C, nitrogen (N) and sulfur
(S) by Elementar Macro Cube, plant-available N as ammonium-nitrogen ($NH_4$-N) and nitrate-
nitrogen ($NO_3$-N) with 2M KCl quantified on Lachat as colorimetric reaction, pH, specific
conductivity, gravimetric water content, bulk density, and porosity. Molecular characterization
included ultra-high resolution C characterization using FTICR-MS and microbial community
assembly analyses using amplicon-based 16S rRNA gene sequencing.

**2.2 FTICR-MS solvent extraction and data acquisition**
Soil organic compounds were extracted using a sequential extraction protocol with polar {water
($H_2O$)} and non-polar {chloroform ($CHCl_3$) and methanol ($CH_3OH$)} solvents per standardized
protocols (Graham et al., 2017a; Tfaily et al., 2015, 2017). Briefly, extracts were prepared by
adding 5 ml of MilliQ $H_2O$ to 5 g of each of the replicate samples in sterile polypropylene
centrifuge tubes (Genesee Scientific, San Diego, USA) suitable for organic solvent extractions
and shaking for 2 h on a Thermo Scientific LP Vortex Mixer. Samples were removed from the
shaker and centrifuged for 5 minutes at 6000 rpm, and the supernatant was removed into a fresh



centrifuge tube. This step was repeated two more times, with the 15 ml supernatant pooled for
each sample and stored at -80 °C until further processing. Next, Folch extraction with $CHCl_3$ and
$CH_3OH$ was performed for each soil pellet left over from the water extraction. Folch extraction
entailed adding 2 ml $CH_3OH$, vortexing for 5 seconds, adding 4 ml $CHCl_3$, vortexing for 5
seconds, followed by of 0.25 ml of MilliQ $H_2O$. The samples were shaken for 1 hr and another
1.25 ml MilliQ $H_2O$ was added and left overnight at 4 °C to obtain bi-layer separation of upper
(polar) layer and the lower (non-polar) layer. The extracts were stored in glass vials at -20 °C
until ready to be used. The water soluble organic carbon (WSOC) fraction was further purified
using a sequential phase extraction protocol to remove salts as per Dittmar et al., 2008. For the
purpose of this study, purified WSOC and $CHCl_3$ fractions were used.  The extracts were
injected into a 12 Tesla Bruker SolariX FTICR-MS located at Environmental Molecular Sciences
Laboratory (EMSL) in Richland, WA, USA. Detailed methods for instrument calibration,
experimental conditions, and data acquisition are provided in Graham et al., 2017a and Tfaily et
al., 2017.

**2.3 FTICR-MS Data Processing**
One hundred forty-four individual scans were averaged for each sample and internally calibrated
using an organic matter homologous series separated by 14 Da ($–CH_2$ groups). The mass
measurement accuracy was less than 1 ppm for singly charged ions across a broad m/z range
(100 - 900 m/z).  Data Analysis software (Bruker Daltonik version 4.2) was used to convert raw
spectra to a list of m/z values applying FTMS peak picker module with a signal-to-noise ratio
(S/N) threshold set to 7 and absolute intensity threshold to the default value of 100. Chemical
formulae were then assigned using in-house software following the Compound Identification
Algorithm, proposed by Kujawinski and Behn (2006), modified by Minor et al. (2012), and





described in Tolić et al. (2017). Peaks below 200 and above 900 were dropped to select only for
calibrated and assigned peaks. Chemical formulae were assigned based on the following criteria:
S/N >7, and mass measurement error < 0.5 ppm, taking into consideration the presence of C, H,
O, N, S, P, and excluding other elements. Detected peaks and associated molecular formula were
uploaded to the in-house pipeline FTICR R Exploratory Data Analysis (FREDA) to obtain: (i)
NOSC values from elemental composition of the organic compounds(Koch and Dittmar, 2006,
2016), (ii) thermodynamic favorability of the compounds calculated as Gibbs Free Energy for the
oxidation half reactions of the organic compounds ($\Delta G^0_{Cox}$) based on the equation $\Delta G^0_{Cox}$ =
60.3-28.5*NOSC (LaRowe and Van Cappellen, 2011), where a higher $\Delta G^0_{Cox}$ indicates a less
thermodynamically favorable species than a lower value (LaRowe and Van Cappellen, 2011),
(iii) abundance of compounds grouped into elemental groups (CHO, CHOS, CHOP, CHNOS,
CHNO, CHNOP, CHOSP, and CHNOSP), and (iv) abundance of compound classes
(carbohydrate-, lipid-, protein-, amino sugar-, lignin-, tannin-, condensed hydrocarbon-, and
unsaturated hydrocarbon-like) based on molar H:C and O:C ratios of the compounds (Bailey et
al., 2017).

Biochemical transformations potentially occurring in each sample were inferred from the
FTICR-MS data by comparing mass differences in peaks within each sample to precise mass
differences for commonly observed  biochemical transformations (Breitling et al., 2006; Stegen
et al., 2018b). The ultra-high mass accuracy of FTICR-MS allows precise mass differences to be
counted for the number of times each transformation was observed within each sample. Briefly,
the mass difference between m/z peaks extracted from each spectrum were compared to
commonly observed mass differences associated with 92 common biochemical transformations



provided in previous publications (Graham et al., 2017a; Stegen et al., 2018c). All possible
pairwise mass differences were calculated within each extraction type for each sample. For
example, a mass difference of 97.05 corresponds to a gain or loss of the amino acid proline,
while a difference of 43.98 corresponds to the gain or loss of a carboxylate molecule.

**2.4 Ecological Modeling**
Null modeling was used to estimate influences of ecological processes on microbial community
composition from rarefied (10000) 16S rRNA amplicon-dependent microbial community
composition and phylogenetic relatedness. The extraction, purification, and sequencing of soil
microbial DNA were performed according to published protocol (Bottos et al., 2018). Sequence
pre-processing, operational taxonomic unit (OTU) table construction and phylogenetic tree
building were performed using an in-house pipeline, HUNDO (Brown et al., 2018). Null
modeling was performed as described previously (Stegen et al., 2013, 2015) with a total of 35
samples to estimate relative influences of deterministic and stochastic selection processes.
Briefly, samples that passed quality control and rarefaction threshold were evaluated for pairwise
phylogenetic turnover between communities, calculated as the difference between the mean-
nearest-taxon-distance ($\beta$MNTD) metric and mean of the null distribution in units of standard
deviation. The difference was significant if the $\beta$-nearest taxon index ($\beta$NTI) $> 2$ or $<-2$
signifying variable or homogenous selection, respectively.
Comparisons within the null distribution ($2 > \beta$NTI$> -2$) represent stochastic processes including
homogenizing dispersal and dispersal limitation or undominated processes. These processes were
evaluated using the Raup-Crick metric extended to account for species relative abundances
(RCbray)(Stegen et al., 2013, 2015).  Homogenizing dispersal was inferred if deviations were
$2 > \beta$NTI$> -2$ and $RC_{bray} < -0.95$, while deviations $2 > \beta$NTI$> -2$ and $RC_{bray} > 0.95$ suggested dispersal



limitation. Undominated processes were represented by comparison within the null distribution
of both metrics ($2 > \beta NTI > -2$ and $0.95 > RC_{bray} > -0.95$). Raw sequences are archived at NCBI
(BioProject PRJNA541992) with reviewer link
(https://dataview.ncbi.nlm.nih.gov/object/PRJNA541992?reviewer=b55qu29emsinvk3udb2rmuf
fqh).

**2.5 Statistical Methods**
Samples were separately analyzed for WSOC and $CHCl_3$ fractions. Within each solvent fraction,
samples were grouped into shallow or deep depths. FTICR-MS dependent metrics including
$\Delta G^0_{Cox}$, and relative abundance of compound classes, total transformations, nitrogen-containing
transformations, and organic nitrogen containing compounds were regressed against specific
conductivity. Regressions were considered significant if $R^2 \geq 0.50$ and $p \leq 0.05$. The
transformation profiles were also regressed with the community assembly processes to determine
the relationship between deterministic/stochastic processes and organic compound
transformations. Mantel tests were used to evaluate similarity between BNTI matrix and
Sorensen matrix of peak presence/absence. The Sorensen distance matrices of WSOC and $CHCl_3$
peaks were regressed against measured variables (soil physicochemical properties) and
community assembly process-variables to determine correlations. Finally, a redundancy analysis
–based stepwise model building with forward model choice was performed to determine
variation in the Hellinger-transformed water-fraction peaks and $CHCl_3$ fraction peaks as
explained by explanatory variables (which included measured soil variables, modeled
community assembly variables, and categorical variables depth and location). All statistical
analyses were performed in the statistical computing language R version 3.5.3 (R Development
Core Team, 2019).




## 3. Results

**3.1 Soil characterization.** The percent of total soil C (%C) in the shallow soils (26.3 $\pm$ 8.3%)

was higher than the deeper soils (4.0 $\pm$ 1.3%) for the lowland soils (i.e. "floodplain" and "inland"

sites), while the upland site had an average %C of 7.4 $\pm$ 0.27% at 10 cm and 2.13 $\pm$ 0.06% at 30

cm (Table S2). No significant relation was observed between %C in the shallow inland and

floodplain soils along the salinity gradient. The percent of total soil N (%N) of the shallow soils

were higher (1.5 $\pm$ 0.40%) than the deeper soils (0.4 $\pm$ 0.08%) for the lowland soils and co-varied

with %C ($r^2$=0.95). The pH of all soils were acidic (5.64 $\pm$ 0.70). The concentrations of $NH_4$-N

and $NO_3$-N showed a consistent trend where $NH_4$-N was 1-2 orders of magnitude higher than

$NO_3$-N in all samples. The specific conductivity (used as a measurement of salinity in this study)

of the shallow soils ranged from 206-866 ($\pm$12)  $\mu S$ $cm^{-1}$ in the lowland soils to 43$\pm$5 $\mu S$ $cm^{-1}$ in

the terrestrial end-member site. The deep soils exhibited specific conductivity ranging from to

148-524 ($\pm$11) $\mu S$ $cm^{-1}$ in the lowland soils to 29.2 $\pm$8 $\mu S$ $cm^{-1}$ in the terrestrial end-member site.

Texture analysis revealed a broad range of sand (4.1 – 40 %), silt (21.4 – 57.9%), and clay (28.6

– 64.8%) fractions.

**3.2 Thermodynamics, compound classes, and elemental composition.**  The calculated $\Delta G^0_{Cox}$

WSOC (Table S3)in shallow soils was consistent with our hypothesis of decreasing

thermodynamic favorability with increasing conductivity. Average $\Delta G^0_{Cox}$ ranged from 53-71 kJ

mol $C^{-1}$ ($R^2$= 0.78, p <0.00001), while no significant relationship between $\Delta G^0_{Cox}$ and specific

conductivity was observed for WSOC fraction in the deeper soils (averaging 51-54 kJ mol $C^{-1}$)

for the floodplain and inland samples (Fig. 2).  The upland site had significantly higher average

$\Delta G^0_{Cox}$ (67-70 kJ mol $C^{-1}$) than the soils near the lowland. The $\Delta G^0_{Cox}$ ($CHCl_3$) at both depths



(Table S4) was higher than the water fractions (ranging between 96-105 kJ mol C$^{-1}$) but did not
show significant relationship with respect to specific conductivity.

Peak profiles for each solvent extraction showed distinct compound classes in the van Krevelen
space, with peaks assigned to specific compound classes according to rules outlined in Kim et
al., 2003 and modified by Bailey et al., 2017. The WSOC fraction was dominated by compounds
classified as protein-, amino sugar-, lignin-, condensed hydrocarbon-, carbohydrate-, and tannin-
like compounds (Table 1), while the CHCl$_3$ fraction had relative high abundances (75% and
higher) of lipid-like compounds (data not shown). A modest percentage of peaks (11-17%) did
not have classes assigned. Unique and common peaks extracted in the WSOC fraction in samples
grouped according to their landscape position and depth [four sites in the floodplain (BC2, BC3,
BC12, and BC13), two sites inland (BC4 and BC14), and one upland site (BC15)] are
represented as H/C to O/C ratio of the compounds (p = 0.05) in Fig. S1.
The shallow WSOC in the floodplain had greater relative abundance of unique lipid (28%)- and
protein (25%)-like compounds with relatively high H:C and low O:C ratios as compared to the
upland site (BC15), which had an 31%, 30%, and 19%  unique peaks representing lignin-,
tannin-, and carbohydrate-like compounds respectively. About 23% of peaks were common in
both groups, including lignin- and condensed hydrocarbon-like compounds (Fig. S1a). Between
the floodplain and the inland samples, high H:C and low O:C ratios representing 47% lipid-,
38% protein-, and 22% amino sugar-like peaks were uniquely present in the floodplain samples
(Fig. S1b). The inland shallow soils had 19% unique higher H:C peaks representing condensed
hydrocarbon-like compounds compared to 1.2% in the upland soil , though most of the
compound classes were observed at both locations (Fig. S1c). Linear regression with specific
conductivity profiles showed significant positive correlation with amino sugar-, protein-, lipid-,





and unsaturated hydrocarbon-like compounds, while condensed hydrocarbon-like compounds
were significantly negatively correlated (Table S5)

For the deep soils, the upland site had 32% unique peaks with relatively high H:C ratios and low
O:C ratios commonly associated with unsaturated hydrocarbon-like compounds, as compared to
the 0.7% in the floodplain which had higher prevalence of unique peaks representing condensed
hydrocarbon (36%)-, and tannin-like (35%) compounds (Table 1, Fig. S1d). The floodplain vs
inland samples had thrice as many unique peaks with high H:C and low O:C ratios representing
lipid-like compounds in the floodplain samples. Comparisons between inland and upland end-
member samples revealed 43% and 37% unique peaks representing low H:C and high O: C ratio
hydrocarbon- and  tannin-like compounds respectively in inland samples, while 32%, 14% 9%,
and 12% of unique peaks were matched to unsaturated hydrocarbon-, lipid-, protein-, and amino
sugar-like compounds respectively in the latter (Table 1, Fig. S1e, f). No significant relationship
was observed with specific conductivity (Table S5). For the $CHCl_3$ fraction, peaks of lipid-like
and unsaturated hydrocarbon-like compounds were observed to be common in all samples (data
not shown) and regressions against specific conductivity were not significant for the compound
classes.

Compositional differences of the organic compounds showed variable heteroatom abundances,
with cumulative heteroatom abundance decreasing with increasing salinity ($R^2$=0.43, p = 0.009)
for shallow fraction of the WSOC.  For the WSOC fraction, heteroatom abundance of CHOP ($R^2$
= 0.61) and CHNOP ($R^2$ = 0.50) containing compounds was consistent with our hypothesis and
significantly (p < 0.01) increased, while CHNOS ($R^2$ = 0.66), and CHNOSP ($R^2$ = 0.62)
abundances were inconsistent with our hypothesis and significantly decreased with increasing
specific conductivity. The elemental composition of the WSOC compounds for deep soils did not
show any significant trend with respect to conductivity. For the $CHCl_3$ fraction, relative
abundance of CHNOP in the shallow soils significantly decreased with specific conductivity ($R^2$
$= 0.57$, $p < 0.01$).

**3.3 Transformation profiles.** In contrast to our expectations, the number of transformations
decreased with increasing salinity in the water fraction of shallow soils ($R^2 = 0.60$, $p < 0.01$) (Fig.
3a, Table S3). We also evaluated N-containing transformations and the abundance of N-
containing compounds in the system. Total nitrogen-containing transformations also decreased
significantly with conductivity but the correlation was not as strong ($R^2 = 0.40$, $p < 0.01$). Total N
containing compounds (Fig. 3b, Table S3) as well as their relative abundance decreased
significantly ($R^2 = 0.74$, $p < 0.01$), with increasing conductivity in the shallow soils for water
fraction.

**3.4 Ecological processes impacting community composition**
Null modeling revealed that community assembly processes were influenced by variable
selection ($\beta NTI > 2$), homogenous selection ($\beta NTI < -2$), dispersal limitation ($2 > \beta NTI > -2$ and
$RC_{bray} > 0.95$), homogenizing dispersal ($2 > \beta NTI > -2$ and $RC_{bray} < -0.95$), and undominated
processes ($2 > \beta NTI > -2$ and $0.95 > RC_{bray} > -0.95$) (Fig. 4 ). Dispersal limitation had the greatest
influence, responsible for 54% of the variation in community composition. The lowest signal
was for homogenizing dispersal (1%), and the signal for homogenous selection (23%) was higher
than variable selection (9%). Together, deterministic processes (variable selection plus
homogeneous selection) were responsible for 32% of the variation in community composition,
with 55% contributed by stochastic processes (dispersal limitation plus homogenizing dispersal).



Variation not accounted by dispersal or selection (i.e., influenced by a mixture of processes)
accounted for the remaining signal (23%). Consistent with influences from both stochastic and
deterministic processes, βNTI relationships with environmental variables were significant (p <
0.05 by Mantel test), but relatively weak (r=0.46 for pH and r=0.31 for specific conductivity)
(Fig. S2).

To evaluate associations between microbial community assembly processes and chemistry,
process estimates were regressed against features of the organic C profiles. Deterministic
processes decreased (Fig S3a) while community assembly processes influenced by non-
deterministic processes increased with increasing number of transformations of organic
compounds (Fig. S3b), although no strong relationships were observed between assembly
processes and transformations (p = 0.027, $R^2$ = 0.11 for deterministic/non-deterministic
processes, p = 0.475, $R^2$ = 0.015 for variable selection, p = 0.054, $R^2$ = 0.10 for homogenous
selection, p = 0.514, $R^2$ = 0.013 for dispersal limitation, and p = 0.627, $R^2$ = 0.007 for
homogenizing dispersal). No significant relationships were observed between assembly
processes and the number of N-containing transformations. Sorensen dissimilarity values based
on the detected FTICR peaks for the water fraction were correlated with distance matrices of
measured environmental variables and estimates of community assembly processes. Weak
positive correlations were observed with $NH_4$-N (r = 0.28), pH (r = 0.27), specific conductivity (r
= 0.41), $NO_3$-N, silt, and clay (r = 0.30) while for the $CHCl_3$ fraction, weak positive correlations
were observed with specific conductivity and $NO_3$-N (r = 0.26) (Fig. S4).  A Mantel test of
FTICR Sorensen dissimilarity vs βNTI values yielded a significant relationship (r = 0.213, p =
0.003) for water fraction but not for $CHCl_3$ fraction (r=0.076, p = 0.152). The stepwise model
building yielded a combination of five variables that were weakly associated with the





composition of water fraction peaks (p=0.026, adj. $R^2$ = 0.217), including sand, dispersal
limitation, $NH_4$-N concentration, specific conductivity, and location. The model explaining
variation in the composition of $CHCl_3$ fraction peaks was non-significant (p = 0.1, adj. $R^2$ =

457 0.05).


**4. DISCUSSION**
Sea level rise is increasing the inland extent of tides and exacerbating storm surge, resulting in
greater salinity intrusion and altered ecosystem behavior across coastal TAIs (Conrads and
Darby, 2017; Ensign and Noe, 2018; Langston et al., 2017; McCarthy et al., 2018; Neubauer et
al., 2013). Site-driven variations in the responses of bulk soil biogeochemical processes (i.e., gas
flux and DOC release) to elevated salinity suggests potentially important influences of
underlying features such as C chemistry and microbial communities. To provide a foundation for
understanding the role of C chemistry and microbial communities on biogeochemical cycling in
coastal soils, we evaluated associations among a landscape-scale soil salinity gradient,
molecular-level soil carbon chemistry, and microbial community assembly processes in order to
ultimately inform future improvements for predictive models. In soils associated with a coastal
first-order drainage basin, we observed salinity-associated gradients in soil organic carbon
character that were not associated with microbial community assembly processes. Our results are
consistent with C chemistry being driven by a combination of spatially-structured inputs and
salinity-associated metabolic responses of microbial communities that are independent of
microbial community composition.

**4.1 Molecular characterization reveals chemical gradients not seen in the bulk C pool**





The systematic shifts observed in the molecular signatures compared to non-significant changes
in bulk C chemistry shows that molecular-level investigations are particularly relevant to
process-based resolution of C biogeochemistry. The absence of bulk C signals mimicking
molecular C signals parallel studies indicating rapid change in molecular constituents of the soil
C pool with no change in gross C content (Graham et al., 2018; Reynolds et al., 2018). A faster
turnover time of C has been observed in microbial biomass as compared to bulk soil organic
matter (Kramer and Gleixner, 2008), which is likely to impact microbe-mediated biochemical C
transformations and lead to chemically complex heterogeneous C signatures likely to be missed
in bulk analysis (Tfaily et al., 2015). The systematic C character shifts exhibited by samples at
the shallow depth suggests that organic C compound pools in shallower soil depths are sensitive
to salinity gradients while deeper depth signatures do not vary systematically across the
landscape. The landscape gradient observed in the shallow soils is likely influenced by a
combination of reduced litterfall from the recently suffering trees, changing understory
vegetation, and algae-rich particulate OM deposition during inundation events that presumably
initiated after the recent culvert removal. In contrast, the deeper soil depths were more similar to
older organo-mineral complexed C in terrestrial soils across various ecosystems and land uses
(Conant et al., 2011; Dungait et al., 2012; Jobbágy and Jackson, 2000; Kramer and Gleixner,
2006, 2008). The lack of any systematic gradients in the mineral-associated soil C provides
further evidence in support of these interpretations, in addition to previous studies showing
mineral-associated soil C to be less responsive to environmental forcings, relative to water
soluble C (Reynolds et al., 2018).

**4.2 Decreases in organic C thermodynamic favorability may restrict microbial activity**



Consistent with our first hypothesis, systematic changes in soil organic C character were
observed with thermodynamically less favorable C present at high salinity in shallow soils. This
gradient was expected to emerge from increased microbial activity at higher salinity leaving
behind less favorable organic C. However, decreases in the number of inferred biochemical
transformations and heteroatom abundances with increasing salinity suggests that microbial
activity decreased with increasing salinity. While difficult to infer direction of causality, these
patterns suggest that less favorable C at higher salinities may constrain microbial activity,
leading to fewer biochemical transformations of the organic C. Thermodynamic limitation of
organic C transformation is likely due to anaerobic conditions (LaRowe and Van Cappellen,
2011), which are indicated by high-moisture content of soils, high $NH_4$-N, and low $NO_3$-N.
Anaerobic conditions restrict oxidation of C compounds based on thermodynamic properties
(i.e., NOSC and $\Delta G^0_{Cox}$) (Boye et al., 2017), and our data suggest that this has the potential to
lead to lower microbial activity in conditions with less favorable organic C.
**4.3 Compound class landscape gradients suggest influences of spatially structured inputs**
Similar to patterns in C thermodynamic favorability, C compound classes showed significant
heterogeneity in shallow soils but had conserved characteristics in deeper soils. The lipid-like
peaks observed in the shallow floodplain samples suggest marine-associated algal-derived lipid
organic matter similar to results observed by Ward et al., 2019 in a coastal wetland setting. In
contrast, lignin-like signatures in the upland site suggest terrestrially derived OM, as has been
observed in other environments where terrestrially-derived organic molecules have a high
abundance of vascular-plant derived material such as lignin (Hedges and Oades, 1997; Ward et
al., 2013). These characteristics also align with reports of saturated soil environments (*e.g.,*
floodplains) exhibiting greater abundance of less-oxygenated organic matter than aerobic
environments (*e.g.,* upland soils) as reported by Tfaily et al., 2014  in organic matter



transformation of a peat column. Our observed landscape gradients in compound class
composition indicate spatially structured inputs of organic C such as particulate OM deposition
(Langley et al., 2007). Combining this outcome with gradients observed in the total number of
biochemical transformations and the contribution of heteroatoms suggests that sources of C
(marine vs terrestrial) and *in situ* processing combine to influence landscape-scale gradients
molecular-level organic C chemistry.

**4.4 Ecological assembly processes are weakly associated with organic C**
Our results show that microbial community assembly is driven by a combination of dispersal
limitation (a stochastic process) and deterministic selection most likely associated with pH, as is
often observed in soils (Fierer, 2017; Fierer and Jackson, 2006; Garbeva et al., 2004). In contrast,
variation in organic C character was associated primarily with specific conductivity. This
suggests that the composition of microbial communities is not mechanistically related to C
chemistry. Consistent with this inference, we found a very weak association between βNTI and
organic C characteristics. Furthermore, and contrary to our hypothesis, we observed a weak
negative association between the influence of deterministic processes and the number of organic
C transformations.

Relatively fast dynamics of organic C compounds compared to relatively slow changes in
microbial composition may underlie the lack of association between assembly processes and C
chemistry (Bramucci et al., 2013). Supporting this interpretation, a recent study evaluating
microbial community composition and C biogeochemistry of soils in a mesohaline marsh
following saltwater intrusion reported immediate changes in C mineralization rates with delayed
shifts in microbial community composition (Dang et al., 2019). Similarly, a 17-year dryland soil





transplant experiment showed large shifts in microbial activity with no change in community
composition (Bond-Lamberty et al., 2016). Furthermore, studies across diverse systems show
disconnect in function and composition. For example, C chemistry and not microbial community
structure or gene expression was found to significantly influence freshwater hyporheic zone
organic matter processing (Graham et al., 2018); environmental conditions influenced the
distribution of functional groups, but not taxonomic composition of marine bacterial and
archaeal communities (Lima-Mendez et al., 2015; Louca et al., 2016); and dynamic community
shifts did not impact functional stability of a methanogenic reactor (Fernández et al., 1999).
Combining our study with these previous investigations provides evidence that soil microbial
community composition does not strongly influence biogeochemical function.

In our system, lack of an association between microbial composition and organic C chemistry is
also likely due to a strong influence of stochastic community assembly. Our null modeling
indicated that dispersal limitation was responsible for 54% of variation in community
composition. Dispersal limitation influences composition by restricting the movement of
organisms through space. Restricted movement enhances the influences of stochastic ecological
drift, which arises through birth and death events that are randomly distributed across taxa
(Green et al., 2004, 2008; Hubbell, 2001; Martiny et al., 2006; McClain et al., 2012; Stegen et
al., 2015). This can result in functionally redundant taxa across the landscape (Loreau, 2004).
Moreover, one can argue as per Louca et al., 2018 that in an open system with regular exposure
to external inputs (e.g., via tides), functional redundancy is expected to occur and lead to a
decoupling of microbial structure and function (Burke et al., 2011; Liebold and Chase, 2017;
Nemergut et al., 2013b).



**Conclusions**

Our results have revealed landscape scale gradients in soil C chemistry in a coastal forested

floodplain, but also show that such gradients are different across soil depths and OC fractions—

occurring only in the shallow, water soluble C pool. In addition, we found little evidence of an

association between C chemistry and microbial community assembly processes, likely due to a

dominant influence of stochastic community assembly (as indicated by a strong influence of

dispersal limitation). We propose that the disconnect between C chemistry and microbial

communities is enhanced by differences in the time scales for which C chemistry and microbial

community composition shift.

Our findings suggest that cross-system heterogeneity observed in coastal soil biogeochemical

responses to salinity are likely associated with molecular-level C chemistry and microbial

physiological responses that are contingent on historical conditions (Goldman et al., 2017;

Hawkes and Keitt, 2015; Hawkes et al., 2017; Stegen et al., 2018a). We further suggest that

microbial community composition may not strongly influence biogeochemical function in

coastal soils. Processes associated with molecular-level C chemistry dynamics are therefore

likely to be a critical component of ecosystem responses to changing salinity dynamics in coastal

TAIs. A full elucidation of these processes will lay a foundation for the development of

mechanistic models of coastal TAI biogeochemical dynamics, providing an opportunity for

better representation of these ecosystems in local, regional, and Earth system models.

**Code and data availability**

Raw sequence data has been uploaded to the National Center for Biotechnology Information's

(NCBI) Sequence Read Archive (SRA) under BioProject PRJNA541992. All other data files and



codes will be uploaded to the Department of Energy's (DOE) Environmental Systems Science
Data Infrastructure for a Virtual Ecosystem (ESS-DIVE) upon manuscript acceptance. Original
codes for community assembly metric calculation are available at Stegen_etal_ISME 2013
github repository https://github.com/stegen/Stegen_etal_ISME_2013.

**Author contribution**
AS designed the study, performed the experiments, conducted data analyses and interpretation,
and wrote the original draft. JI and CG collected the samples and created site maps. MTF, RKC,
and JT provided input on FTICR methodology, conducted the FTICR-MS instrument run, and
handled quality filtering and pre-processing of FTICR scans. VLB and NDW contributed to
funding acquisition, site selection, study design conceptualization, interpretation of results and
editing. JCS contributed to funding acquisition, study design conceptualization, interpretation of
results, reviewing and editing. All authors provided feedback on the manuscript.

**Competing interests**
The authors declare no conflict of interest.

**Acknowledgements**
This work is part of the PREMIS Initiative at Pacific Northwest National Laboratory (PNNL). It
was conducted under the Laboratory Directed Research and Development Program at PNNL, a
multi-program national laboratory operated by Battelle for the U.S. Department of Energy under



Contract DE-AC05-76RL01830. A portion of the research was performed using EMSL
(grid.436923.9), a DOE Office of Science User Facility sponsored by the Office of Biological
and Environmental Research. We would like to acknowledge Yuliya Farris and Sarah Fansler for
DNA extraction and sequencing respectively, Colin Brislawn for processing amplicon-sequence
data, and the Central Analytical Laboratory at Oregon State University for conducting soil
chemical analysis.

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

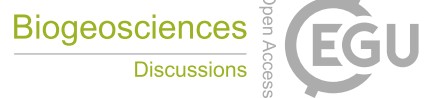

**Table 1.** Relative peak abundances (%) of compound classes in the water extracted organic
carbon fraction averaged across replicates per site. Samples are ordered according to their depth
profile (shallow and deep) and their relative position in the landscape: floodplain (Fp), inland (I),
and upland (U). Abbreviations: Con HC (condensed hydrocarbon), UnsatHC (unsaturated
hydrocarbon), Other (no classification assigned)

| Site/Depth | Landscape position | Protein | Amino Sugar | Lipid | Lignin | Con HC | Tannin | Other | Carb | Unsat HC |
|---|---|---|---|---|---|---|---|---|---|---|
| BC2_Shallow | Fp | 17.2 | 3.3 | 9.4 | 31.0 | 22.3 | 13.2 | 0.5 | 1.8 | 1.3 |
| BC3_Shallow | Fp | 21.6 | 3.8 | 11.5 | 27.3 | 23.0 | 9.8 | 0.4 | 1.5 | 1.2 |
| BC4_Shallow | I | 1.6 | 0.6 | 0.3 | 45.3 | 32.2 | 18.9 | 0.04 | 0.8 | 0.2 |
| BC12_Shallow | Fp | 7.6 | 1.8 | 4.0 | 38.1 | 31.2 | 15.3 | 0.1 | 1.2 | 0.7 |
| BC13_Shallow | Fp | 13.3 | 2.6 | 5.9 | 33.4 | 28.6 | 14.4 | 0.2 | 0.9 | 1.0 |
| BC14_Shallow | I | 6.1 | 1.7 | 1.6 | 37.0 | 35.8 | 16. | 0.2 | 0.8 | 0.5 |
| BC15_Shallow | U | 3.7 | 1.5 | 1.3 | 51.8 | 18.5 | 21.0 | 0.2 | 1.5 | 0.5 |
| BC2_Deep | Fp | 2.3 | 0.5 | 1.5 | 41.2 | 27.2 | 25.7 | 0.2 | 1.1 | 0.3 |
| BC3_Deep | Fp | 3.2 | 0.3 | 3.1 | 34.1 | 33.4 | 24.4 | 0.3 | 0.9 | 0.2 |
| BC4_Deep | I | 2.8 | 0.8 | 0.6 | 50.4 | 27.7 | 16.5 | 0.2 | 0.7 | 0.2 |
| BC12_Deep | Fp | 2.29 | 0.40 | 1.43 | 43.3 | 27.9 | 22.9 | 0.2 | 1.2 | 0.3 |
| BC13_Deep | Fp | 3.47 | 0.62 | 2.00 | 39.8 | 33.6 | 19.2 | 0.2 | 0.8 | 0.3 |
| BC14_Deep | I | 1.71 | 0.76 | 0.57 | 43.7 | 32.5 | 19.34 | 0.2 | 1.0 | 0.2 |
| BC15_Deep | U | 9.51 | 2.55 | 4.70 | 63.8 | 5.1 | 9.93 | 0.7 | 1.0 | 2.6 |


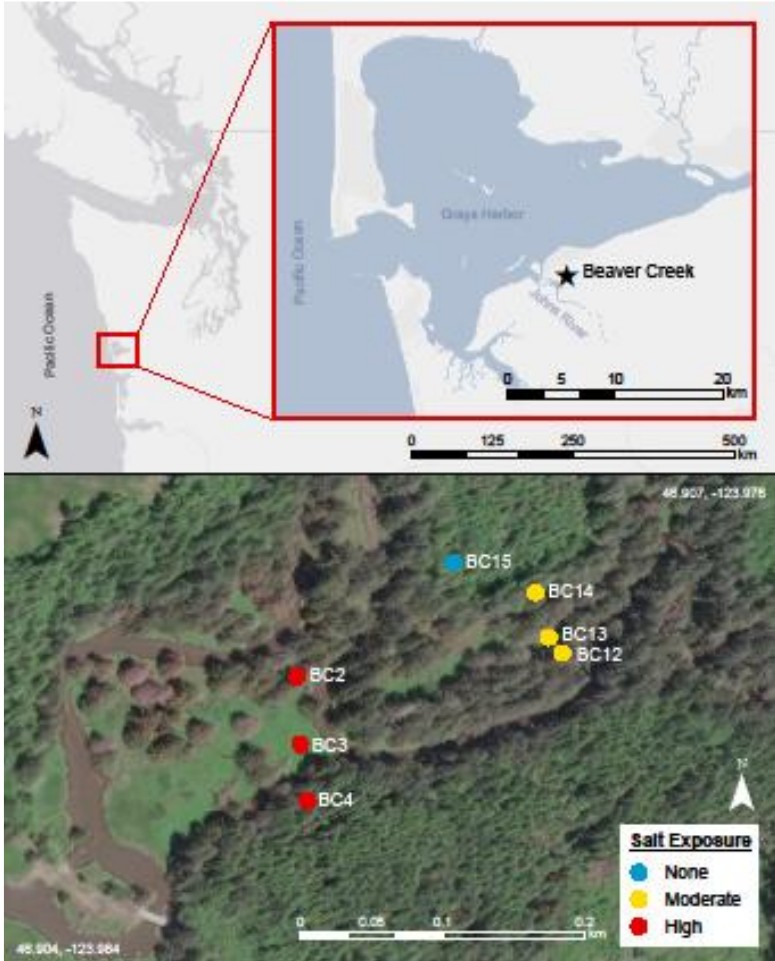

**Figure 1**. Study site Beaver Creek in the Olympic Peninsula in western Washington. The creek

is a first order stream with tidal exchange restored in 2014. Top panel shows site location in

western Washington with inset panel zoomed in to show site close to Johns River. Bottom panel

shows soil sampling locations at the high salt exposure (BC2, BC3, BC4) transect, moderate salt

exposure (BC12, BC13, BC14) transect, and terrestrial upland (BC15) site. The transects with

six sampling sites experience periodic inundation episodes which result in surface pooling of

tidal water.  Map was created using ArcGIS 10.5 software (ESRI, 2017). Coordinate System:

GCS WGS 1984.





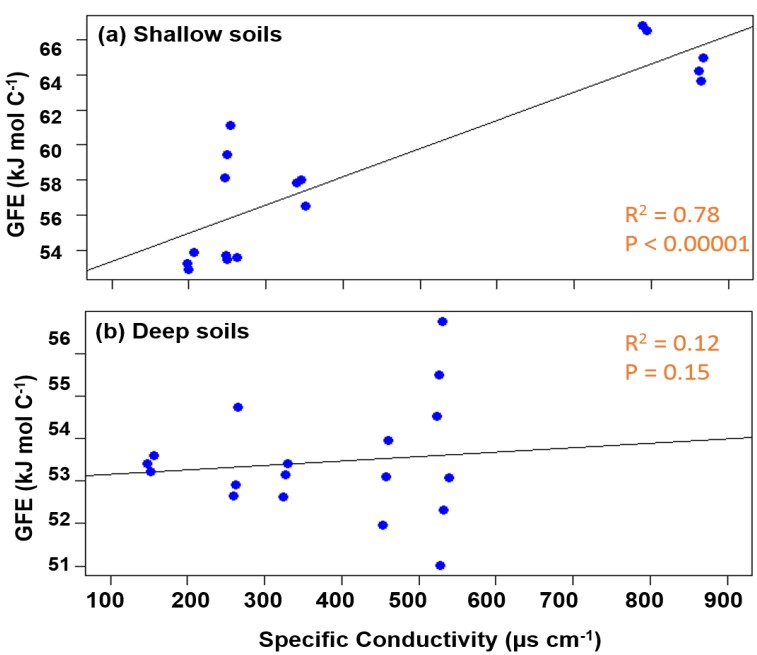


**Figure 2.** Average Gibbs Free Energy (GFE) of samples in the water fraction of shallow soils

increased with increasing specific conductivity (a) while no change was observed in the deeper

soils (b).





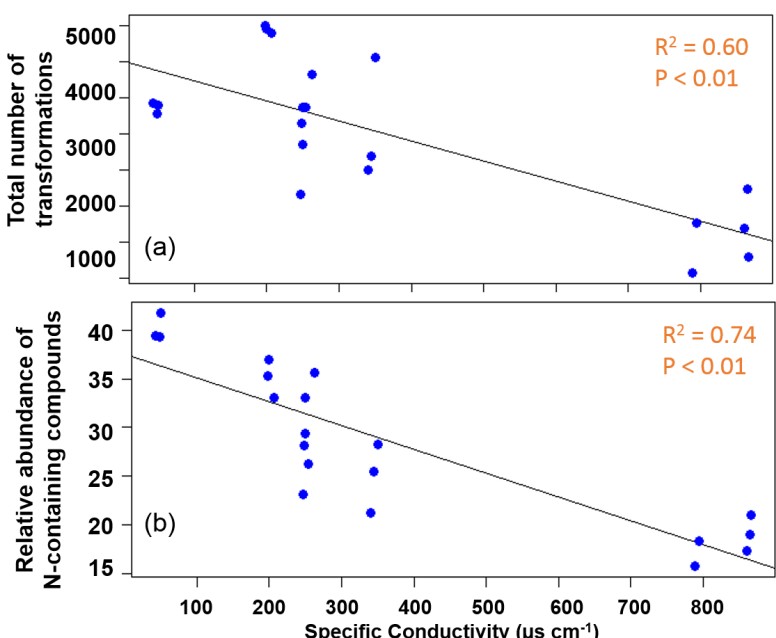


**Figure 3.** The total number of inferred transformations (a) and total abundance of N-containing

compounds (b) in the water fraction of shallow soils show significant negative correlations with

increasing specific conductivity. No significant relationships were observed for water fraction of

deeper soils or for the CHCl$_3$ fraction in shallow or deeper soils.





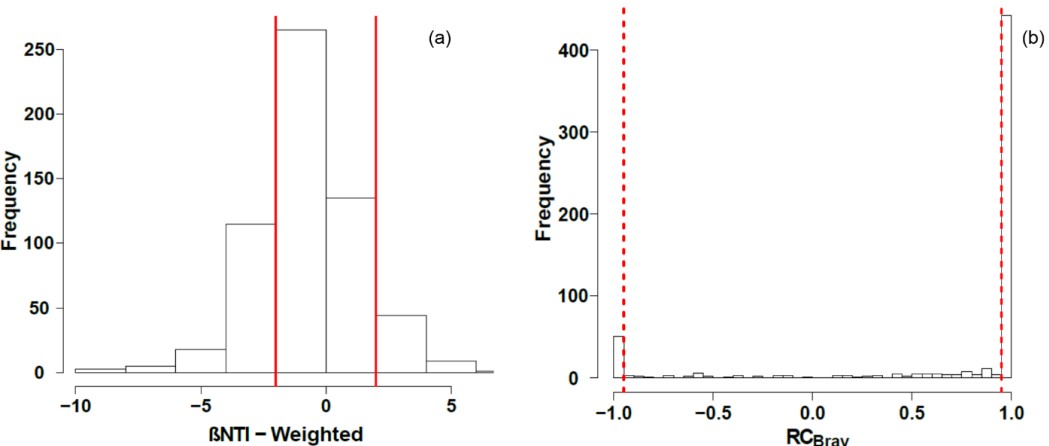


**Figure 4.** Histograms representing the observed distribution of comparisons based on (a) Beta-

near taxon index (βNTI) and (b) Raup Crick metric (RC$_{Bray}$). Red lines represent the significance

thresholds, whereby values outside their bounds are significantly different from the null

distribution.

1084