# Peer review of "Spatial gradients in soil-carbon character of a coastal forested floodplain are associated with abiotic features, but not microbial communities"

_Biogeosciences, 2019_

## Referee Comment (RC1) · Anonymous Referee #1 · 20 Jun 2019

This paper attempts to identify associations between soil carbon chemistry (molecular composition of SOC fractions revealed by FT-ICR MS analysis) and microbial communities (analyzed by 16S rRNA) at the coastal terrestrial-aquatic interfaces (TAIs) influenced by salinity gradients along a small first order stream in the Washington Coast. These two high-resolution techniques generate tons of information on organic matter chemistry and microbial community composition, which allows detailed examination of their linkages. The introduction part nicely lays out the rationale and hypothesis of this study and the paper is overall well written. However, there are a few issues that need

to be addressed.

First of all, the extracted fractions and analyzed molecules are only a small part of the SOC, which may (very likely) not reflect the overall chemistry of total soil organic matter. In this regard, the title and related descriptions should be clarified—it is "chemical characteristics of soil carbon fractions" instead of "soil-carbon character". It should also be mentioned in the Methods how much SOC was extracted by the employed method. Given the lability of WSOC, it is hence more likely to be influenced by microbial decomposition compared to bulk SOC, but it is also strongly influenced by direct inputs of low-molecular compounds from root exudates, etc.—this brings my second point.

Despite the nicely formulated hypotheses for this paper, the authors seem to largely ignore (or underestimate) the influence of input processes on the molecular composition of extractable OC. Water- and solvent-extractable OC may derive from direct plant and algal inputs other than depolymerization of soil macromolecules by microbial-mediated enzyme attack. How would root exudates contribute to the thermodynamically less favorable C, for instance? Do you have an estimate of NPP (hence soil inputs) along the study gradient? The observed changes in C chemistry may well be a combined result of decomposition and input processes. Similarly, how would photo-oxidation affect the signal?

Regarding the analysis and interpretation of the FT-ICR MS data, I am not convinced that the number of common/unique formulas is the best parameter to describe changes in OC chemistry. The relative abundance of these formulas should be considered. How representative are the unique formulas in the overall abundance of total MS peaks, for instance? How does the relative abundance of common formulas change with salinity gradient? Hemingway et al. 2017 GCA give a good example for such kind of analysis.

Specific comments: Line 219: Why these two depths? Line 395: Relationship with what?

---

## Referee Comment (RC2) · Anonymous Referee #2 · 20 Jun 2019

**General Comments**

This study investigated effects of salinity in coastal forested floodplains on soil carbon pools and microbial community structure. The authors use FTIR to characterize the chemical species within the soil C pool and molecular techniques to characterize and correlate microbial community structure to soil C chemistry, as well as compare all measurements between the different salinity sites. The ecosystems studied are unique and interesting and at the fringe of TIAs which have clear importance as sea levels continue to rise and salt water intrusion into freshwater systems is likely to alter soil and ecosystem level C cycling dynamics within these fringe ecosystems. I think the study has value to be published and readers of BGC will be interested in the findings, although I have a few major suggestions, primarily in the writing style.

I find the writing to be good overall, but is too generalized in that there is not enough detail given for the use of specific terminology, particularly in the introduction but also throughout the manuscript. This is especially important to reach a broad enough audience and make this research have higher impact. For instance, microbial biochemical transformations, or biogeochemical transformations, were terms used a lot but it is not clear which transformations or processes the authors are referring too. See more comments on that below. Further, I found that although the hypotheses were introduced in the introduction, the lack of specificity in the introduction regarding each hypotheses made it challenging to follow the authors' logic. Overall, I think the authors should write the introduction with more specific examples from the literature they site, showing the gaps in knowledge on the subject (salinity effects on soil processes in TAIs) and how this study addresses those gaps by asking specific hypotheses.

**Abstract**

Abstract is too vague, making it hard to follow what the authors studied, measured, and how to interpret these results.

L26 TAI doesn't really need an acronym here because it is never used in the abstract again.

L31 Heteroatom seems like a very specific term. It would be helpful to know the definition of a heteroatom or to use a more common term.

L34 please state salinity range here or previously

L34 what does inferred biochemical transformations mean? Are these the ones that were measured? It would be more direct to just state which biochemical transformations are being referred to.

L35 which metrics of microbial activity were measured?

L41 "Null modelling revealed strong influences on dispersal limitation" I am unclear what this means. So the microbial communities were spatially variable or distinct from each other depending on where the samples were taken?

L44 What is a community assembly process? Does this just mean C mineralization, or nitrification, or some other microbially driven process?

L44 "lack of an association" can the authors be more specific. How were microbial communities measured? PLFA? Molecular techniques? Which part of the microbial communities were compared to C chemistry?

L44 "C chemistry" can the authors be more specific? Which C compounds?

L45 "disconnect btn community and C biogeochem" can you be more specific? What part of the community and biogeochemical processes were disconnected?

**Introduction**

L100 change rates to processes. Rates are not microbially driven, processes are. Which rates/processes are decoupled? Which gas fluxes? CO2 and CH4?

L101 Size of C pool… is this referring to the concentration of DOC mentioned in L100? Clarify

L100-103 How does a decoupling between the C pool size and microbial activity in saline environments suggest it is due to salinity exposure history? Based on how this paragraph is written, it seems like the authors can only say it is due to elevated salinity. Clarify what is meant by salinity exposure history.

L107 Microbial-activity driven??? Needs to be reworded

L98-120 this paragraph starts about discussion between relationships (or lack of) between gas fluxes, DOC, and microbes and ends in a discussion about methods for analyzing chemical constituents of SOC. This should be split up into two paragraphs or reworded to provide better flow. Maybe the first part can be incorporated into the previous paragraph.

L135 please define heteroatom as it is not necessarily a common term when describing SOC

L137-138 What is it about increasing salinity that leads to greater heteroatom concentration? This point is unsupported by the first part of the sentence which seems to just be a general statement.

L140 N mining…please be more specific…N uptake from soil? In the form of inorganic or organic N? Is it already available for uptake or do the microbes secrete enzymes to liberate organically bound N in order to take up inorganic N?

L143 clarify that the flooding that results in marine derived OM is flooding from marine salt water terrestrial systems. I assume the terrestrial ecosystem is freshwater, but up to this point there has been no mention of whether the flooded environment is already saline or is freshwater.

L150-165 As a reader, I am having trouble following the logic of this paragraph mainly due to the lack of specificity in the use of terms such as community assembly processes, ecological assembly processes, biogeochemical processes, deterministic and stochastic assembly processes, and dispersal processes. Can the authors give examples of what processes they are specifically referring too? It is too general to build a hypothesis off of based on salinity changes in the environment. What is the difference between a community and ecological assembly process? And which can be grouped into deterministic and stochastic categories?

L160 Why subsurface microbial ecology? Are the effects different in soil surface horizons?

**Methods**

L184 provide lat and long coordinates at the end of the first sentence

L186 Can any information be provided on the extent of inundation onto the landscape? Or the size of the floodplain?

L189 define psu

L197-199 please provide common names for species as well

L204-207 How long were the transects? At what distances along the transects were samples taken?

L208-209 I prefer to see soil taxonomic information as well as soil series information. It gives readers a choice on what to interpret. I am not that familiar with Ocosta or Mopang soil series so it provides very little information to me about the soil characteristics without having to go look it up on the NRCS.

L210 Any idea on water table depth? How deep is the water that pools on the surface?

L217-218 It would be nice to know the elevation of the floodplain, inland, and upland transects.

L219 Are shallow samples 0-10 cm depth?

L224-229 There should be a little bit more detail here on each method, or maybe citations to the methods used at the very least. Provide make, model, company etc. for Lachat. How was pH measured, conductivity, GWC, BD, and porosity?!?! What about pre processing? Was large organic matter removed including roots and litter, or retained. Were samples air dried, sieved, etc..?

L227-229 this doesn't need to be included here. It is in the following sections.

L243 followed by of….check wording

L294-295 It seems like more information should be provided on the microbial DNA procedures.

**Results**

L352 Table S3 is almost unreadable in the small font size

L392 missing comma after 14%

Why have the authors chosen to not include any taxonomical data on the microbial communities? It seems that this would be very useful information and I assume this information was obtainable from the methods used.

**Discussion**

L463-464 Here the authors have at least provided some examples of the biogeochem processes they are interested referring to.

L471 characteristics?

L472 Authors mention spatially structure inputs. I assume this is in reference to land scape variation but it would be helpful to be more specific.

L473 What metabolic responses of microbial communities were measured in this study?

L489 Suffering….awkward wording….Also this appears to be the first mention of forests/tress under stress. Can the authors elaborate on this or provide site level data confirming this?

L494 The authors didn't measure mineral associated C. How then can comments be made about that fraction of the soil C pool? Maybe because these are generally silt and clay rich soils compared to the clearly much more organic surface soils?

L533 How did the authors determine dispersal limitation? Does this mean that the microbial communities were different between the sites? This would not be surprising but is hard to determine since microbial taxonomic structure was not provided.

L542 relatively fast dynamics…..unclear what this means….fast changes in the chemistry of the C? be specific.

L556-557 I find this statement to be highly speculative given the one sampling date and the lack of measurements of any actual microbial activity metrics. I would argue that there were no measures of biogeochemical functioning in this study, just measures of the outcome of biogeochemical processes (e.g. remaining C compounds, N compounds etc.). L159 is a more accurate statement…..microbial community (although I think the microbial community structure, abundance of different taxonomic groups, etc. should be shown) was compared to soil C chemistry.

L562-563 This is the first time, as far as I can tell, that the authors attempted to define dispersal limitation. This information needs to be given when this is first mentioned in the manuscript.

L563-L566 How does restrictive movement of microbial communities in space lead to functional redundancy? It seems like this would actually reduce functional redundancy as spatially restricted microbial communities become more specialized over time especially in salinated and non salinated soils which likely has a marked effect on the microbial community structure.

**Tables and Graphs**

Figure 1. It would be helpful to have a label for the waterway in the right hand side of the bottom panel. I think that is Beaver Creek but unsure. Maybe this tributary to Johns River does not have a name though?

Figure 2 and 3. I recommend color coding the points for each of the three sites so readers can see where they fall out on the regression line.

Table S3 font size should be increased if possible

---

## Author Comment (AC1) · 11 Jul 2019

Response to Referee #1 comments. These are also provided as a supplement pdf file.

1. This paper attempts to identify associations between soil carbon chemistry (molecular composition of SOC fractions revealed by FT-ICR MS analysis) and microbial communities (analyzed by 16S rRNA) at the coastal terrestrial-aquatic interfaces (TAIs) influenced by salinity gradients along a small first order stream in the Washington Coast. These two high-resolution techniques generate tons of information on organic matter

chemistry and microbial community composition, which allows detailed examination of their linkages. The introduction part nicely lays out the rationale and hypothesis of this study and the paper is overall well written. However, there are a few issues that need to be addressed.

We appreciate that the reviewer recognizes the value in the data we report. We have carefully considered all of the review comments and have provided responses.

2. First of all, the extracted fractions and analyzed molecules are only a small part of the SOC, which may (very likely) not reflect the overall chemistry of total soil organic matter. In this regard, the title and related descriptions should be clarifiedâAËŸTit is "chemical ËĞ characteristics of soil carbon fractions" instead of "soil-carbon character".

We will edit the title in the revised version to indicate this change and clarify in the text that we use "soil carbon character" in our text to indicate chemical characteristics of soil carbon fractions.

3. It should also be mentioned in the Methods how much SOC was extracted by the employed method.

The sequential extraction protocol is able to extract 2-15% of total organic carbon as per previous established protocols (Tfaily et al., 2015, 2017). We will note this in the methods. The goal here is to get a representative sample of the water soluble and chloroform soluble pool, which the two references cited above prove. This is a well-established protocol and we are confident that the extractions represent both polar- and non-polar soil organic carbon fractions.

4. Given the lability of WSOC, it is hence more likely to be influenced by microbial decomposition compared to bulk SOC, but it is also strongly influenced by direct inputs of low-molecular compounds from root exudates, etc.âAËŸTthis brings my second point. ËĞ Despite the nicely formulated hypotheses for this paper, the authors seem to largely ignore (or underestimate) the influence of input processes on the molecular composition of extractable OC. Water- and solvent-extractable OC may derive from direct plant and algal inputs other than depolymerization of soil macromolecules by microbial-mediated enzyme attack. How would root exudates contribute to the thermodynamically less favorable C, for instance? Do you have an estimate of NPP (hence soil inputs) along the study gradient? The observed changes in C chemistry may well be a combined result of decomposition and input processes. Similarly, how would photo-oxidation affect the signal?

Agreed that extractable OC is driven by inputs (plant and algal derived) and that the observed changes in C chemistry is a combined result of decomposition/input processes which we cannot separate out. We will add sentences in the introduction to indicate these.

While we agree that root exudates may impact the carbon signatures, this was not the focus of our study. However, we attempted to evaluate common root exudate composition from literature so that we could derive Gibbs Free Energy of the compounds, particularly focusing on compounds associated with vegetation found at our transects, but were unable to find any relevant information. We thank the reviewer for the suggestion and it indeed will be an interesting new study to see how root exudate chemistry varies across the salinity gradient.

Unfortunately, we do not have a good estimate of NPP for the field site at this time. Using MODIS NPP product is also not a viable option because MODIS is 1 km pixel scale while the Beaver Creek site itself is 3.8 km2. However, we are in the process of collecting data to make such calculations for future studies focused on plant physiology at this same site. In the future we plan to examine changes in soil carbon chemistry as the floodplain soils becomes increasingly saline, and will include NPP information in our future efforts. Thank you for the recommendation.

We do not anticipate photo-oxidation at 10 cm and 19-30 cm soil depths.

5. Regarding the analysis and interpretation of the FT-ICR MS data, I am not convinced

that the number of common/unique formulas is the best parameter to describe changes in OC chemistry.

We have limited our interpretation of common/unique formulas only from the perspective of similarity to compound classes at different sampling locations. We do not claim this to be the best parameter. We have looked at other features to describe changes in OC chemistry including Gibbs Free Energy, heteroatom content, and inferred biochemical transformations.

6. The relative abundance of these formulas should be considered.

We have provided relative peak abundances of compound classes in the water extracted organic carbon fraction (Table 1, Line 1048).

7. How representative are the unique formulas in the overall abundance of total MS peaks, for instance? How does the relative abundance of common formulas change with salinity gradient? Hemingway et al. 2017 GCA give a good example for such kind of analysis.

This is an interesting idea suggested by the reviewer but it is beyond the scope of our study. The analysis being suggested here is different from what we asked and evaluated. We did pairwise comparisons by grouping samples according to landscape position and depth (Lines 367-370), with common/unique features comparable between groups like Floodplain versus Inland, Floodplain versus Terrestrial, and Inland versus Terrestrial at two individual depths. Comparing sample 1 to sample 2, and then sample 1 to sample 3, and so on to evaluate how common formulas change with salinity gradient will lead to results that will be difficult to interpret because the commonality/unique features as a fraction of common/unique peaks is not a property inherent to sample, but only emerges when compared to samples. Therefore, we did not evaluate representativeness of unique formulas in the overall peaks because the unique/common feature is relative and dependent on which groups are being compared. We specifically used the common/unique formulas to understand relative compound class similarity at

transects, irrespective of the salinity gradient. Given that our system is in transition with variable tidal/terrestrial inputs, we used the common/unique features to understand the predominant compound class patterns in our field site.

8. Specific comments: Line 219: Why these two depths?

The two soil depths were chosen based on visual soil characteristics. The shallow depth was the organic-rich horizon, while the deeper depth was characterized by lighter colored, clay-rich soils. We did not go any deeper due to logistical constraints—during the time of sampling, the holes back-filled with water up to roughly the depth of the "deep" samples. The depth of distinct layers were fairly consistent across all floodplain sites and not as evident in the upland forest site, however we maintained consistency with how the floodplain soils were collected in this case.

9. Line 395: Relationship with what?

Relationship of compound-like peak abundances with specific conductivity. We will edit the manuscript to reflect this change.

Please also note the supplement to this comment:
https://www.biogeosciences-discuss.net/bg-2019-193/bg-2019-193-AC1-supplement.pdf

---

## Author Comment (AC2) · 11 Jul 2019

**Responses to Referee #2 comments are in blue**

**Anonymous Referee #2

**General Comments**
1.  This study investigated effects of salinity in coastal forested floodplains on soil carbon pools and microbial community structure. The authors use FTIR to characterize the chemical species within the soil C pool and molecular techniques to characterize and correlate microbial community structure to soil C chemistry, as well as compare all measurements between the different salinity sites.

    We used FTICR-MS (Fourier Transform Ion Cyclotron Mass Spectrometry) and not FTIR (Fourier Transform Infrared Spectroscopy). FTICR-MS is a mass analysis that determines mass-to-charge ratio of ions based on cyclotron frequency of ionized compounds in a fixed magnetic field, and therefore allows us to evaluate ultra high-resolution profiling of organic compounds from perspectives of thermodynamics, inferred biochemical transformations, and similarity to organic compound classes. FTIR measures infrared absorption and emission spectra and does not provide a mass-to-charge ratio of ions.

2.  The ecosystems studied are unique and interesting and at the fringe of TIAs which have clear importance as sea levels continue to rise and salt water intrusion into freshwater systems is likely to alter soil and ecosystem level C cycling dynamics within these fringe ecosystems. I think the study has value to be published and readers of BGC will be interested in the findings, although I have a few major suggestions, primarily in the writing style.

    *We appreciate that the reviewer recognizes the value in the research and data we report. We have carefully considered all of the review comments and have provided detailed responses*

3.  I find the writing to be good overall, but is too generalized in that there is not enough detail given for the use of specific terminology, particularly in the introduction but also throughout the manuscript.

    *We thank Reviewer 2 for their constructive comments and feedback. We will provide definition of terminologies and/or refer readers to relevant citations that discuss the terminologies in detail in the revised version.*

4.  This is especially important to reach a broad enough audience and make this research have higher impact. For instance, microbial biochemical transformations, or biogeochemical transformations, were terms used a lot but it is not clear which transformations or processes the authors are referring too. See more comments on that below.

    *The transformations refer to biochemical transformations that were potentially occurring within each sample. For which transformations we are referring to, please see lines 279-289*

*and* (Breitling et al., 2006; Stegen et al., 2018) *which highlight the commonly observed biochemical transformations. The ultra-high mass accuracy of FTICR-MS allows us to infer these transformations.*

5.  Further, I found that although the hypotheses were introduced in the introduction, the lack of specificity in the introduction regarding each hypotheses made it challenging to follow the authors' logic.

    *Our introduction lays out the expectations based on literature review, setting up the stage for our hypotheses in lines 167-173. However, for clarity, we will add sentences in the relevant Introduction paragraphs to link to the specific hypothesis listed lines 167-173.*

6.  Overall, I think the authors should write the introduction with more specific examples from the literature they site, showing the gaps in knowledge on the subject (salinity effects on soil processes in TAIs) and how this study addresses those gaps by asking specific hypotheses.

    *Relevant examples from literature are provided in lines.* ***Salinity effects on soil processes as they relate to gas flux, dissolved organic carbon, and bulk carbon*** *in TAIs are discussed in lines 76-95 and 105-108,* ***gaps are discussed*** *in lines 98-108, 115-116, 152-155 and how our* ***study addresses those gaps*** *by testing specific hypotheses in lines 123-150, 166-167. We will carefully review and attempt to clarify more examples in the revised version.*

**Abstract**
7.  Abstract is too vague, making it hard to follow what the authors studied, measured, and how to interpret these results.

    *The abstract has been written for a general audience, capturing the essence of our analyses, results, and interpretation. However, we will rephrase certain sections of the abstract to convey a succinct message. To reiterate, Lines 30-32, and 40 demonstrate what we* ***studied and measured*** *(salinity associated shift in organic C and associated microbial community assembly processes),* ***what we analyzed*** *(organic C thermodynamics, biochemical transformations, heteroatom content, relationship between microbial community assembly processes and C chemistry), and* ***the interpretation*** *(lines 42-48).*

8.  L26 TAI doesn't really need an acronym here because it is never used in the abstract again.

    *It is used in Line 28 and therefore the acronym is justified.*

9.  L31Heteroatom seems like a very specific term. It would be helpful to know the definition of a heteroatom or to use a more common term.

    *Added N-,S-,P- containing heteroatom. We have also explained what a heteroatom is in lines 135-136.*

10. L34 please state salinity range here or previously

   *Will add in revised version.*

11. L34 what does inferred biochemical transformations mean? Are these the ones that were measured? It would be more direct to just state which biochemical transformations are being referred to.

   *These biochemical transformations mean gain or loss of molecules based on gain or loss of mass in the spectra. These transformations can only be inferred (and not measured) from the FTICR-MS data by matching the mass differences in peaks to mass transformations. For example, a mass difference of 99.07 corresponds to gain or loss of the amino acid valine while a difference of 179.06 corresponds to gain or loss of a glucose molecule. We have provided this explanation in the methods (line 279-289). There are 92 common biochemical transformations (based on commonly observed mass difference associated with biochemical transformations which we evaluated for in our data, and referenced from the following: (Bailey et al., 2017; Breitling et al., 2006; Graham et al., 2017a, 2018; Stegen et al., 2018). We will add the phrase "based on mass differences" in line 34 for clarity.*

12. L35 which metrics of microbial activity were measured?

   *We do not claim to measure microbial activity. We state that decreasing thermodynamic favorability, biochemical transformations, and heteroatom content imply less favorable organic carbon accumulation which in turn indicates lower microbial activity at higher salinity.*

13. L41 "Null modelling revealed strong influences on dispersal limitation" I am unclear what this means. So the microbial communities were spatially variable or distinct from each other depending on where the samples were taken?

   *Strong influences of dispersal limitation influence microbial community composition by restricting the movement of organisms through space, which suggests that communities are distinct from each other but their assembly is governed by stochastic processes (ecological drift arising through birth and death events that are randomly distributed across taxa). We will edit the revised manuscript to reflect this is simpler terms.*

14. L44 What is a community assembly process? Does this just mean C mineralization, or nitrification, or some other microbially driven process?

   *Community assembly process is defined as the process governing composition of ecological communities (Stegen et al., 2012, 2013, 2015). Therefore, here it is the assembly of microbial communities. These are ecologically governed and driven by either deterministic processes (selection resulting from different organisms having different levels of fitness for a given set of environmental conditions including abiotic variables and biotic factors related to organismal interactions) or stochastic processes (random birth/death events, drift). We will edit the revised manuscript to reflect a succinct message.*

15. L44 "lack of an association" can the authors be more specific. How were microbial communities measured? PLFA? Molecular techniques? Which part of the microbial communities were compared to C chemistry?

*There was no significant relationship between community assembly metrics (BNTI matrix which indicates phylogenetic relatedness of the samples, variable selection, homogeneous selection, dispersal limitation, and homogenizing dispersal) with C chemistry. The microbial community composition was determined using 16S rRNA amplicon sequencing approach, the microbial community assembly metrics were calculated using the Null modeling approach which used the 16S rRNA amplicon-sequence data derived composition information and phylogenetic tree.*
*The community assembly metrics were compared to C chemistry. We will rephrase this sentence to better reflect the message.*

16. L44 "C chemistry" can the authors be more specific? Which C compounds?

*Associations were not evaluated with C compounds. They were evaluated with C chemistry information including Gibbs Free energy, transformation profiles, and heteroatom content, and peak information (as written in methods section; lines 321-334).*

17. L45 "disconnect btn community and C biogeochem" can you be more specific? What part of the community and biogeochemical processes were disconnected?

*The community assembly process variables were disconnected from C chemistry. The community composition (βNTI) relationship with environmental variables were significant (p < 0.05 by Mantel test), but relatively weak (r=0.46 for pH and r=0.31 for specific conductivity), which was not true for Gibb's free energy which was strongly related to specific conductivity trends. We did not find any significant associations between community assembly process variables and C chemistry including Gibbs Free energy, transformation profiles and mass spectra peaks. These explanations are part of results section (Lines 435-460).*

**Introduction**
18. L100 change rates to processes. Rates are not microbially driven, processes are. Which rates/processes are decoupled? Which gas fluxes? CO2 and CH4?

*We will edit the line in the revised manuscript.*

*Microbial-driven carbon metabolism rates are decoupled from dissolved organic carbon concentrations.*

*We are referring to both $CO_2$ and $CH_4$ flux trends and will restructure the sentence to reflect the same in the revised version.*

19. L101 Size of C pool… is this referring to the concentration of DOC mentioned in L100? Clarify

    *Yes.*

20. L100-103 How does a decoupling between the C pool size and microbial activity in saline environments suggest it is due to salinity exposure history? Based on how this paragraph is written, it seems like the authors can only say it is due to elevated salinity. Clarify what is meant by salinity exposure history.

    *We derive inference from our literature review in the previous paragraph that shows a general trend where soils from historically fresh environments when exposed to increasing salinity show increase in $CO_2$ flux while soils in saline environments exposed to increasing salinity show reduced $CO_2$ flux. This signifies that salinity exposure history is critical. We then go on to explain the implications of the salinity exposure history on resource environment of microbes, bulk C signatures that cannot represent molecular-level changes, and often no observable shifts in bulk C even when a salinity exposure occurs in historically saline or fresh environment. We will rephrase Line 100 to read "Relatively consistent gas flux responses **with salinity exposure history….**"*

21. L107 Microbial-activity driven??? Needs to be reworded

    *We do not agree. Microbial activity drives carbon cycling.*

22. L98-120 this paragraph starts about discussion between relationships (or lack of) between gas fluxes, DOC, and microbes and ends in a discussion about methods for analyzing chemical constituents of SOC. This should be split up into two paragraphs or reworded to provide better flow. Maybe the first part can be incorporated into the previous paragraph.

    *We will edit this paragraph and move the methods discussion to end of paragraph 4.*

23. L135 please define heteroatom as it is not necessarily a common term when describing SOC

    *We have defined it in the line (organic compounds containing N, S, P).*

24. L137-138 What is it about increasing salinity that leads to greater heteroatom concentration? This point is unsupported by the first part of the sentence which seems to just be a general statement.

    *It is expected that actively growing microbes increase heteroatom containing organic compounds (as indicated by the references cited). Since $CO_2$ fluxes trended to be increasing with increasing salinity in a freshwater system, we hypothesized that as a freshwater system (and changing to saltwater since 2014 after culvert removal), Beaver Creek would also see increasing activity and therefore greater heteroatom content with*

*increasing salinity. We will add a sentence in line 125 to indicate the salinity exposure history of our site and therefore our expectations of ecosystem behavior.*

25. L140 N mining…please be more specific…N uptake from soil? In the form of inorganic or organic N? Is it already available for uptake or do the microbes secrete enzymes to liberate organically bound N in order to take up inorganic N?

*We will edit the sentence to reflect a strategy that may require microbes to breakdown organic molecules to extract N (i.e. N mining).*

26. L143 clarify that the flooding that results in marine derived OM is flooding from marine salt water terrestrial systems. I assume the terrestrial ecosystem is freshwater, but up to this point there has been no mention of whether the flooded environment is already saline or is freshwater.

*We agree and will revise the line to clarify this as tidal flooding. Our site is unique in this sense that it used to be saline until a culver was built to divert water, blocking off tidal inundation. The site therefore turned fresh until 2014 when the culvert was removed and tidal activity resumed. We agree that we should mention that the system is freshwater in recent history and will rephrase line 100 to indicate this and therefore provide context to line 143. Further details are provided in Methods lines 190-194.*

27. L150-165 As a reader, I am having trouble following the logic of this paragraph mainly due to the lack of specificity in the use of terms such as community assembly processes, ecological assembly processes, biogeochemical processes, deterministic and stochastic assembly processes, and dispersal processes. Can the authors give examples of what processes they are specifically referring too? It is too general to build a hypothesis off of based on salinity changes in the environment. What is the difference between a community and ecological assembly process? And which can be grouped into deterministic and stochastic categories?

*We will edit the text in the revised version to add clarity. Further, we direct the reviewer to* (Graham et al., 2017a, 2017b, Stegen et al., 2012, 2013, 2015) *references to gain detailed information about community assembly processes. As per our response to Reviewer 2 early in this response document: Community assembly process is defined as the process governing composition of ecological communities (Stegen et al., 2012, 2015). Therefore, here it is the assembly of microbial communities. These are ecologically governed and driven by either deterministic processes (selection resulting from different organisms having different levels of fitness for a given set of environmental conditions including abiotic variables and biotic factors related to organismal interactions) or stochastic processes (random birth/death events, drift).*

28. L160 Why subsurface microbial ecology? Are the effects different in soil surface horizons?

*Edit will be made in the revised manuscript to indicate surface and sub-surface.*

**Methods**

29. L184 provide lat and long coordinates at the end of the first sentence

    *Edit will be made in the revised manuscript.*

30. L186 Can any information be provided on the extent of inundation onto the landscape? Or the size of the floodplain?

    *The Beaver Creek watershed is 3.8 km². The tidal floodplain makes up 0.5 km² of this total watershed area. Information will be added in the revised manuscript.*

31. L189 define psu

    *Edit will be made in the revised manuscript.*

32. L197-199 please provide common names for species as well

    *Agrostis stolinifera (creeping bentgrass), Tsuga heterophylla (Western hemlock), Picea sitchensis (Sitka spruce). Edits will be made in the revised manuscript.*

33. L204-207 How long were the transects? At what distances along the transects were samples taken?

    *Each transect was on roughly 80-90 m. Samples were collected ~35 m apart (information provided in line 216).*

34. L208-209 I prefer to see soil taxonomic information as well as soil series information. It gives readers a choice on what to interpret. I am not that familiar with Ocosta or Mopang soil series so it provides very little information to me about the soil characteristics without having to go look it up on the NRCS.

    *Edit will be made in the revised manuscript. These are Andisols.*

35. L210 Any idea on water table depth? How deep is the water that pools on the surface?

    *The water table depth in the floodplain is variable both seasonally and throughout tidal cycles/flood events. During floods, there can be almost 1 m of standing water depending on the tide height. During the summer we have observed the water table to be deeper than 60 cm below the ground surface, whereas in the winter it is higher (e.g. 20-30 cm). We have learned from a series of piezometer transects installed across the floodplain that the hydrology of this system is very complex. We are working on a 3-D hydrologic model to describe these dynamics along with salinity, but this effort is beyond the scope of the present manuscript, and unfortunately will not be published soon enough to be referenced here. We will provide the following info for context, while attempting to not lean too heavily on unpublished results:*

*"The transects experience periodic inundation episodes which result in surface pooling of tidal water, which can be up to ~1m deep. The water table varies seasonally and during tidal cycles and inundation events, ranging from about 0 to 1m below the ground surface (Ward, unpublished)."*

36. L217-218 It would be nice to know the elevation of the floodplain, inland, and upland transects.

    We will provide the elevation in the revised manuscript.

37. L219 Are shallow samples 0-10 cm depth?

    *Shallow samples were collected at 10 cm depth.*

38. L224-229 There should be a little bit more detail here on each method, or maybe citations to the methods used at the very least. Provide make, model, company etc. for Lachat. How was pH measured, conductivity, GWC, BD, and porosity?!?! What about pre processing? Was large organic matter removed including roots and litter, or retained. Were samples air dried, sieved, etc..?

    *We will add the relevant information in our revised manuscript. We did not have any litter at the depths we collected the soils from. Sieving ensured removal of roots.*

39. L227-229 this doesn't need to be included here. It is in the following sections.

    *Will edit in the revised manuscript.*

40. L243 followed by of….check wording

    *Will edit in the revised manuscript.*

41. L294-295 It seems like more information should be provided on the microbial DNA procedures.

    *All procedures were performed as per* (Bottos et al., 2018) *that has been cited in the manuscript. Without further specifics from the reviewer, we believe details in Bottos et al., 2018 are adequate.*

**Results**
42. L352 Table S3 is almost unreadable in the small font size

    *This is a large file. We can alternately have an excel table uploaded if allowed to do so or have this table be hosted with our data and provide a link.*

43. L392 missing comma after 14%

*Will correct in the revised version.*

44. Why have the authors chosen to not include any taxonomical data on the microbial communities? Itseems that this would be very useful information and I assume this information was obtainable from the methods used.

*We agree that it can be useful to know the taxonomic composition but this information is not central to testing our hypotheses. As such, we will include a bar chart as a supplemental figure, and also provide the OTU table with taxonomic assignments as a supplemental file so that others can more deeply evaluate taxonomic structure. For the response purpose, we have also included a phylum-level bar plot as an example of our supplemental addition to our revised manuscript.*

**Discussion**

45. L463-464 Here the authors have at least provided some examples of the biogeochem processes they are interested referring to.

*Agreed. We have also indicated the same in lines 76-89.*

46. L471 characteristics?

*We will edit in the revised version.*

47. L472 Authors mention spatially structure inputs. I assume this is in reference to land scape variation but it would be helpful to be more specific.

*Yes. We will edit this sentence in the revised manuscript to indicate landscape-variation and therefore differences in type (terrestrial versus aquatic) as well as terrestrial vegetation input differences.*

48. L473 What metabolic responses of microbial communities were measured in this study?

*We did not measure any metabolic response but inferred thermodynamic favorability of organic compound metabolites. We will revise the manuscript to indicate that additional work is needed that focuses on quantifying inputs and measuring microbial metabolism.*

49. L489 Suffering….awkward wording….Also this appears to be the first mention of forests/tress under stress. Can the authors elaborate on this or provide site level data confirming this?

*Edited. We write about trees under stress in the methods section in line 193. We will provide additional reference from our recent vegetation survey publication at this site.*

50. L494 The authors didn't measure mineral associated C. How then can comments be made about that fraction of the soil C pool? Maybe because these are generally silt and clay rich soils compared to the clearly much more organic surface soils?

   *We will edit the methods to indicate that we treated CHCl3-extracted organic carbon as proxy for mineral bound C as per (Graham et al., 2017a).*

51. L533 How did the authors determine dispersal limitation? Does this mean that the microbial communities were different between the sites? This would not be surprising but is hard to determine since microbial taxonomic structure was not provided.

   *The influence of dispersal limitation (relative to other community assembly processes) was quantified using a previously established null modeling approach, as discussed in lines 326-332 (see Stegen et al. 2012, 2013, 2015). We chose to focus on the ecological community assembly processes rather than look directly at taxonomic composition because of previously published simulation model-based predictions indicating a potential association between assembly processes and biogeochemical function. Null modeling is required because examining taxonomic composition directly does not provide information on the ecological processes governing community composition.*

52. L542 relatively fast dynamics…..unclear what this means….fast changes in the chemistry of the C? be specific.

   *Yes, rapid changes in reactions. Will edit in revised manuscript to reflect the change.*

53. L556-557 I find this statement to be highly speculative given the one sampling date and the lack of measurements of any actual microbial activity metrics. I would argue that there were no measures of biogeochemical functioning in this study, just measures of the outcome of biogeochemical processes (e.g. remaining C compounds, N compounds etc.).

   *We recognize that our study captures one time point and we did not measure microbial activity. We are speculative within limits to conclude that community composition in our system does not influence biogeochemical function. We are citing other papers that show poor association between composition and biogeochemical function. That's why the sentence is structured as 'combining our study with these previous…'*

54. L159 is a more accurate statement…..microbial community (although I think the microbial community structure, abundance of different taxonomic groups, etc. should be shown) was compared to soil C chemistry.

   *Line 559 response: we used the BNTI metric to show that microbial community composition and phylogenetic relatedness were not associated with OC. We would also like to guide the reviewer to lines 544 to 555 that show that community composition may not change while function changes. We will also include a supplemental OTU table and bar plot in the revised manuscript to show the taxonomic groups.*

55. L562-563 This is the first time, as far as I can tell, that the authors attempted to define dispersal limitation. This information needs to be given when this is first mentioned in the manuscript.

   *We will add brief description in the Introduction in the revised manuscript with detailed citations to guide readers to resources that explain community assembly processes in detail.*

56. L563-L566 How does restrictive movement of microbial communities in space lead to functional redundancy? It seems like this would actually reduce functional redundancy as spatially restricted microbial communities become more specialized over time especially in salinated and non salinated soils which likely has a marked effect on the microbial community structure.

   *Because ecological drift (enabled by dispersal limitation) can lead to the random loss of taxa within local communities, it can result in different communities containing different, but functionally redundant taxa. We will add this to the revised manuscript version.*

**Tables and Graphs**
57. Figure 1. It would be helpful to have a label for the waterway in the right hand side of the bottom panel. I think that is Beaver Creek but unsure. Maybe this tributary to Johns River does not have a name though?

   *Sorry for the confusion—the waterway in the bottom panel is in fact Beaver Creek. The confluence of Beaver Creek and Johns River is not shown in this panel. We have added a label accordingly. We will edit this as requested in the revised version.*

58. Figure 2 and 3. I recommend color coding the points for each of the three sites so readers can see where they fall out on the regression line.

   *We will make the change in the revised manuscript.*

59. Table S3 font size should be increased if possible

   *This is a large file. We can alternately have an excel table uploaded if allowed to do so or have this table be hosted with our data and provide a link.*

---

## Author Response (AR1)

*Response to comments on "Spatial gradients in soil-carbon character of a coastal forested floodplain are associated with abiotic features, but not microbial communities" by Aditi Sengupta et al.*

Dear Dr. Ji-Hyung Park,

We greatly appreciate your and reviewers' thoughtful assessment of our manuscript "Spatial gradients in soil-carbon character of a coastal forested floodplain are associated with abiotic features, but not microbial communities" [Paper bg-2019-193]. Please see our response document for detailed responses to all your and reviewers' comments, with references to line numbers in the track-changes manuscript where changes can be found.

As a result of these changes we believe this work is significantly strengthened and hope you and the reviewers agree.

Sincerely, for the authors,

Dr. Aditi Sengupta

**Response to Associate Editor comments are in blue.**

The two reviewers recognized the scientific value and novelty of your manuscript. Agreeing with the two reviewers, I am pleased to tell you that you can now submit your revised manuscript to be considered for publication in Biogeosciences. However, given the large number of comments and some important issues raised by reviewers, I have to recommend 'reconsider after major revisiosn.'

*Thank you for considering our manuscript for a revise and resubmit following major revisions. We have carefully considered your and reviewers' comments and have revised the manuscript. We have also responded point-by-point to your and reviewers' comments. Our responses are provided in italicized blue font. The line numbers of the revisions are of the marked-up copy with track changes. Additionally, we have also provided a new conceptual diagram line 682, 1199-1208 that summarizes the linkages revealed in our study among salinity, organic C thermodynamic favorability, and inferred levels of microbial activity.*

1. I found that your indices of "biochemical transformations" based on FTICR-MS analysis might represent a key methodological approach. Although FTICR-MS has recently become a popular and common approach in various biogeochemical research fields, more detailed descriptions of the analytical procedures and accuracy, and its methodological limitations would help readers understand the unique approach employed in your study.

   *Thank you for the suggestions. We have now provided detailed descriptions and methodological limitations as requested in lines 132-134, 266-269, 284-301, and 338-343. We have also added a new figure (Figure 2) lines 338-340, 1164-1169 to help illustrate the mass difference and transformation concept. The following have been added:*

*"An important caveat is that factors such as redox state, physical protection, mineral associations, and microbial community composition can alter this pure chemistry-based expectation (Schmidt et al., 2011)."*

*"Soil organic compounds were extracted using a sequential extraction protocol with polar {water ($H_2O$)} and non-polar {chloroform ($CHCl_3$) (representing mineral-bound fraction)} solvents per standardized protocols (Graham et al., 2017a; Tfaily et al., 2015, 2017), which extract about 2-15% of total organic carbon and represent both polar and non-polar soil organic carbon fractions. Importantly, our analyses do not depend on extracting a large portion of the C found within a given soil sample. Instead, we assume that the extracted fraction is a representative sub-sample."*

*"Briefly, samples were acidified to pH 2 with 85% phosphoric acid. The samples were passed through Bond Elut PPL cartridges (©Agilent Technologies) that were preactivated with $CH_3OH$. The cartridges were washed 5x with 10mM HCl followed by nitrogen-gas drying. Next, 1.5 ml CH3OH, a solvent that is compatible with direct analysis on the FTICR-MS, was used to elute the samples from the cartridge thus avoiding an additional evaporation step that might reduce the chance of losing volatile organic compounds. While SPE by PPL has shown not to be very effective in extracting several major classes of DOM compounds that have high ESI efficiencies, such as carboxylic acids and organo-sulfur compounds, and that out-competed other less functionalized compounds (e.g., carbohydrates) for charge in the ESI source (Tfaily et al., 2012), it is critical for marine and estuary DOM samples as it provides complete desalination of the sample. Loss of small molecules such as simple sugars is known to happen during SPE however this is not a concern for the current study as FTICR-MS is sensitive to compounds above ~200 Da. In this study, SPE by PPL isolated a major DOM fraction, that is salt-free, allowing for DOM characterization by FTICR-MS(Dittmar et al., 2008b). While we didn't measure SPE extraction efficiency for this study, it usually ranges between 40 and 62 % depending on the sample (Dittmar et al. 2008). Samples that are collected from the same ecosystem have shown to have similar extraction efficiency. For the purpose of this study, the WSOC (representing the water soluble fraction) and $CHCl_3$ (representing the mineral-bound fraction) fractions were used."*

*"All possible pairwise mass differences were calculated within each extraction type for each sample. As an example, Figure 2 shows the comparison of two peaks with a mass difference of 2.01586, which indicates a putative hydrogenation reaction between the two organic molecules represented by the associated peaks. It is important to note that direct injection electrospray ionization FTICR-MS cannot distinguish between isomers such as in the case of a mass difference corresponding to a loss of gain of glucose, fructose, or galactose."*

2. First, please provide more details on the procedures of 'sequential phase extraction protocol to remove salts as per Dittmar et al., 2008' (e.g., SPE sorbent, eluent, recovery, etc.) in addition to the recovery of WSOC commented by the first reviewer. Please refer to the papers that have warned the technical limitation of SPE (e.g., Anal Bioanal Chem (2016) 408: 4809-4819)

*Thank you for the suggestions. We have now provided a brief description of the SPE protocol, responded to the comment about recovery of WSOC by reviewer 1 and have also called out the limitation of SPE. The added text and line numbers are provided in response to the previous comment.*

3. Considering the limitation of FTICR-MS as a tool for quantifying molecular peaks, you might also need to discuss the limitation of your approach of comparing mass differences in peaks. For instance, it has been criticized that FTICR-MS results cannot accurately quantify changes in peak intensity between samples from some incubation experiments.

*Direct injection (DI) FTICR MS is known to be a qualitative or semi-quantitative approach and this is mainly due to the use of electrospray ionization. In general, a molecule's relative ionization efficiency is determined by the relative abilities of different functional groups to stabilize a negative charge. In negative ion mode, ESI preferentially ionizes molecules that can carry a negative charge as a result of deprotonation. It is possible to use DI FTICR MS in three ways: 1) considering the presence or absence of a peak in the mass spectrum and 2) MS peak intensities which are a function of the abundance of compound in the extract, the ionization efficiency, and their ability to compete for ionization with other compounds in the extract, and 3) accurate mass differences between compounds which doesn't take into consideration the intensity of the peak. In addition to offering putative identification of formula, FTICR MS, due to its ultra-high resolution, has the potential to identify the connectivity between related metabolites since chemically transformed species will be related by measurable and clearly defined mass differences regardless of their intensity.*

*In this study, we used approaches 1 and 3 and did not use peak intensities since the intensity-dependent approach is known to have issues with charge competition. That is, we used the presence or absence of a peak in the mass spectrum as well as the accurate masses to compare mass differences in peaks. In this approach metabolites whose mass differed by the expected amount (within 1 ppm) were considered to be putatively related by the corresponding metabolic transformation. One potential bias of this approach is that it doesn't take into consideration isomers. For example, a mass difference of $C_6H_{12}O_6$ could be either glucose or fructose or galactose. Using FTICR MS alone we can't differentiate between these three sugars. However, this approach was only used to classify sugar versus non-sugar transformations and therefore distinguishing between the simple sugars was not a goal of this study. It is important to note that approach 3 uses the masses of the peaks from the mass spectrum regardless of whether they are assigned a molecular formula or not.*

*As suggested, we added a sentence to the materials and methods that discusses briefly one of the main limitations of this approach. "It is important to note that the direct injection electrospray ionization FTICR-MS approach cannot distinguish between isomers such as in the case of a mass difference corresponding to a loss of gain of glucose, fructose, or galactose" in lines 340-343.*

4. Second, please think about whether presenting some representative van Krevelen diagrams (as shown in Fig S1) would help readers better figure out your approach of mass difference. You have only four figures in the main manuscript, so one additional figure would not add too much load, I think.

*We have now added a new figure (Figure 2, provided below), lines 338-340, 1164-1169to help illustrate the mass difference and transformation concept. to illustrate the mass difference approach. We consider this illustration more informative than van Krevelen and presents what is meant by mass difference of peaks affiliated with a biochemical transformation.*

[Figure]

**Figure 2.** a) Negative mode FTICR-MS (full spectrum); b) zoom in at ~ 450 m/z showing an example of our FTICR-MS spectra overlain with peak mass assignments (red), and a biochemical transformation (mass difference between peaks, denoted in blue). Y axis denotes peak intensities, X-axis denotes mass-to-charge ratio.

5. I would like to ask you to make all the changes easily identifiable in a marked-up manuscript and a point-by-point reply to all the comments offered by the two reviewers and myself. I would also suggest that you specify the line numbers of the revised parts in your responses to the reviewers' and my own comments.

*We have provided point-by-point response to your and reviewers' comments and have also provided edits in a track-change document. The line numbers provided in this response document correspond to the track-changed manuscript.*

**Responses to Referee #1 comments are in blue**

**Anonymous Referee #1
1. This paper attempts to identify associations between soil carbon chemistry (molecular composition of SOC fractions revealed by FT-ICR MS analysis) and microbial communities (analyzed by 16S rRNA) at the coastal terrestrial-aquatic interfaces (TAIs) influenced by salinity gradients along a small first order stream in the Washington Coast. These two high-resolution techniques generate tons of information on organic matter chemistry and microbial community composition, which allows detailed examination of their linkages. The introduction part nicely lays out the rationale and hypothesis of this study and the paper is overall well written. However, there are a few issues that need to be addressed.

*We appreciate that the reviewer recognizes the value in the data we report. We have carefully considered all the review comments and have provided responses.*

2. First of all, the extracted fractions and analyzed molecules are only a small part of the SOC, which may (very likely) not reflect the overall chemistry of total soil organic matter. In this regard, the title and related descriptions should be clarified it is "chemical characteristics of soil carbon fractions" instead of "soil-carbon character".

*We have edited the title in the revised version to indicate this change and clarify in the text that we use "soil carbon character" in our text to indicate chemical characteristics of soil carbon fractions. For example, in line 558 we state that "we observed salinity-associated gradients in soil organic carbon fractions that were not associated with microbial community assembly processes", line 579-580 and 594-595 "in chemical characteristics of soil carbon fractions",*

3. It should also be mentioned in the Methods how much SOC was extracted by the employed method.

*The following has been added in lines 266-271: "..which extract about 2-15% of total organic carbon and represent both polar and non-polar soil organic carbon fractions. Importantly, our analyses do not depend on extracting a large portion of the C found within a given soil sample. Instead, we assume that the extracted fraction is a representative sub-sample. This is a standard approach and assumption made in any study examining metabolites or other types of organic molecules in soil."*

4. Given the lability of WSOC, it is hence more likely to be influenced by microbial decomposition compared to bulk SOC, but it is also strongly influenced by direct inputs of low-molecular compounds from root exudates, etc. This brings my second point. Despite the nicely formulated hypotheses for this paper, the authors seem to largely ignore (or underestimate) the influence of input processes on the molecular composition of extractable OC. Water- and solvent-extractable OC may derive from direct plant and algal inputs other than depolymerization of soil macromolecules by microbial-mediated enzyme attack. How would root exudates contribute to the thermodynamically less favorable C, for instance? Do you have an estimate of NPP (hence soil inputs) along the study gradient? The observed changes in C chemistry may well be a combined result of decomposition and input processes. Similarly, how would photo-oxidation affect the signal?

*Agreed that extractable OC is influenced by inputs (plant and algal derived) and that the observed changes in C chemistry are a combined result of decomposition/input processes which we cannot separate out. We have added sentences in the Introduction (lines72-74) "While multiple processes impact TAI carbon pools (e.g., tidal-inputs, in situ root exudates and litter inputs, decomposition processes), there is some indication that microbial diversity and composition impact soil C storage and mineralization (Mau et al., 2015; Trivedi et al., 2016)." We also now discuss this in lines 566-568 as a caveat to indicate these multiple factors. "Future work should also use tools like Nuclear Magnetic Resonance and Gas Chromatograph-Mass Spectrometry to evaluate how low molecular weight OC (like those contributed by root exudates) varies with salinity."*

*While we agree that root exudates may impact the carbon signatures, this was not the focus of our study. A key reason is that FTICR-MS is not able to detect root exudates that have low molecular weight. This is a missing piece that can be filled in the future with NMR or GC-MS data. We thank the reviewer for the suggestion and it indeed will be an interesting new study to see how root exudate chemistry varies across the salinity gradient.*

*Unfortunately, we do not have a good estimate of NPP for the field site at this time. Using MODIS NPP products is also not a viable option because MODIS is 1 km pixel scale while the Beaver Creek site itself is only 3.8 $km^2$. However, we are in the process of collecting data to make such calculations for future studies focused on plant physiology at this same site. In the future we plan to examine changes in soil carbon chemistry as the floodplain soils become increasingly saline, and will include NPP information in our future efforts. Thank you for the recommendation.*

*We do not anticipate photo-oxidation at 10 cm and 19-30 cm soil depths.*

5. Regarding the analysis and interpretation of the FT-ICR MS data, I am not convinced that the number of common/unique formulas is the best parameter to describe changes in OC chemistry.

*We agree that this is not the best parameter/approach. This is a primary reason we focused much of our study on other features to describe changes in OC chemistry including Gibbs Free Energy, heteroatom content, and inferred biochemical transformations. The common/unique compound classes are a minor component of our analyses to show relative heterogeneity of compound classes between samples.*

6. The relative abundance of these formulas should be considered.

*We believe the reviewer is asking about the relative abundances of compound classes, as opposed to formulas. As such, we have provided relative peak abundances of compound classes in the water extracted organic carbon fraction (Table 1).*

7. How representative are the unique formulas in the overall abundance of total MS peaks, for instance? How does the relative abundance of common formulas change with salinity gradient? Hemingway et al. 2017 GCA give a good example for such kind of analysis.

*This is an interesting idea but it is beyond the scope of our study. The analysis being suggested would be adding additional concepts, questions, and hypotheses. We feel that our study is quite rich already with respect to concepts, questions, and hypotheses that are all linked together into a collective whole. The suggested analysis is intriguing, but doesn't clearly fit into our integrated vision for our study. We would therefore much prefer to explore this analysis in future work.*

*There are also some difficult issues that arise from the suggested analyses, as follows. First, we note that we did pairwise comparisons by grouping samples according to landscape position and depth (Lines 454-456), with common/unique features comparable between groups like Floodplain versus Inland, Floodplain versus Terrestrial, and Inland versus Terrestrial at two individual depths. However, comparing sample 1 to sample 2, and then sample 1 to sample 3, and so on to evaluate how common formulas change across the salinity gradient will lead to results that will be difficult to interpret. This is because the fraction of peaks that are common/unique is not a property inherent to a sample, but only emerges when comparing samples to each other. Therefore, we did not evaluate representativeness of unique formulas in the overall peaks because the unique/common feature are dependent on which groups are being compared. As such, we believe that additional methods development is needed to properly implement the suggested analyses.*

8. Specific comments: Line 219: Why these two depths?

*The two soil depths were chosen based on visual soil characteristics. The shallow depth was the organic-rich horizon, while the deeper depth was characterized by lighter colored, clay-rich soils. We did not go any deeper due to logistical constraints—during the time of sampling, the holes back-filled with water up to roughly the depth of the "deep" samples. The depth of distinct layers were consistent across all floodplain sites, though not as evident in the upland forest site.*

9. Line 395: Relationship with what?

*In line 481-482, the text now reads "No significant relationship between compound-class abundances and specific conductivity was observed (Table S5)."*

**Responses to Referee #2 comments are in blue**

**Anonymous Referee #2

**General Comments**
1. This study investigated effects of salinity in coastal forested floodplains on soil carbon pools and microbial community structure. The authors use FTIR to characterize the chemical species within the soil C pool and molecular techniques to characterize and correlate microbial community structure to soil C chemistry, as well as compare all measurements between the different salinity sites.

   One important detail to note is that we used FTICR-MS (Fourier Transform Ion Cyclotron Mass Spectrometry) and not FTIR (Fourier Transform Infrared Spectroscopy). FTICR-MS quantifies mass-to-charge ratio of ions based on cyclotron frequency of ionized compounds in a fixed magnetic field, and therefore allows us to evaluate ultra-high-resolution profiles of organic compounds from perspectives of thermodynamics, inferred biochemical transformations, and similarity to organic compound classes. FTIR measures infrared absorption and emission spectra and does not provide a mass-to-charge ratio of organic molecules.

2.  The ecosystems studied are unique and interesting and at the fringe of TIAs which have clear importance as sea levels continue to rise and salt water intrusion into freshwater systems is likely to alter soil and ecosystem level C cycling dynamics within these fringe ecosystems. I think the study has value to be published and readers of BGC will be interested in the findings, although I have a few major suggestions, primarily in the writing style.

    *We appreciate that the reviewer recognizes the value in the research and data we report. We have carefully considered all of the reviewer's comments and have provided detailed responses.*

3.  I find the writing to be good overall, but is too generalized in that there is not enough detail given for the use of specific terminology, particularly in the introduction but also throughout the manuscript.

    *We thank Reviewer 2 for their constructive comments and feedback. We have provided definition of terminologies and/or refer readers to relevant citations that discuss the terminologies in detail in the revised version.*

4.  This is especially important to reach a broad enough audience and make this research have higher impact. For instance, microbial biochemical transformations, or biogeochemical transformations, were terms used a lot but it is not clear which transformations or processes the authors are referring too. See more comments on that below.

    *The transformations refer to biochemical transformations that were potentially occurring within each sample. For which transformations we are referring to, please see lines 331-343 and* (Breitling et al., 2006; Stegen et al., 2018) *which highlight the commonly observed biochemical transformations. The ultra-high mass accuracy of FTICR-MS allows us to putatively infer these transformations. We have additionally provided a new figure (Figure 2), line 338-340, 1164-1169to illustrare the mass difference based biochemical transformation approach.*

5.  Further, I found that although the hypotheses were introduced in the introduction, the lack of specificity in the introduction regarding each hypotheses made it challenging to follow the authors' logic.

    *Our introduction lays out the expectations based on literature review, setting the stage for our hypotheses in lines 192-204. For clarity, we have added sentences, for example, in (lines 125-129) "we derived a series of expectations by first recognizing that (1) our study system is a historically freshwater system, only recently being exposed to salt water due to removal of a culvert in 2014 (see Methods), and (2) microbial activity increases with increasing salinity in historically freshwater systems", lines 136-139 "however, we assume that OM*

*reactivity follows NOSC, thereby leading to our first expectation/hypothesis: the average $\Delta G^0_{Cox}$ of OM will increase with increasing salinity as organic compounds with greater thermodynamic favorability are preferentially depleted (LaRowe and Van Cappellen, 2011) due to microbial activity increasing with salinity", 173-176 "combined with evidence of increasing microbial activity with increasing salinity (discussed above) leads to a fifth hypothesis".*

6. Overall, I think the authors should write the introduction with more specific examples from the literature they site, showing the gaps in knowledge on the subject (salinity effects on soil processes in TAIs) and how this study addresses those gaps by asking specific hypotheses.

   *Relevant examples from literature are provided as follows. **Salinity effects on soil processes as they relate to gas flux, dissolved organic carbon, and bulk carbon** in TAIs are discussed in lines 84-107, **gaps are discussed** in lines 103-107, 114-116, 157-160 and how our **study addresses those gaps** by testing specific hypotheses in lines 125-129 and 173-178, 192-204. We have carefully reviewed and attempted to clarify more examples in the revised version in lines 94, 98-100, 115-116.*

**Abstract**
7. Abstract is too vague, making it hard to follow what the authors studied, measured, and how to interpret these results.

   *The abstract has been written for a general audience, capturing the essence of our analyses, results, and interpretation. We have rephrased certain sections of the abstract to convey a succinct message. To reiterate, Lines 30-32, and 40-42 demonstrate what we **studied and measured** (salinity associated shift in organic C and associated microbial community assembly processes), **what we analyzed** (organic C thermodynamics, biochemical transformations, heteroatom content, relationship between microbial community assembly processes and C chemistry), and **the interpretation** (lines 42-48).*

8. L26 TAI doesn't really need an acronym here because it is never used in the abstract again.

   *It is used in the next sentence and therefore the acronym is justified.*

9. L31Heteroatom seems like a very specific term. It would be helpful to know the definition of a heteroatom or to use a more common term.

   *We edited the Abstract text to indicate that heteroatoms are N,S, or P atoms contained within organic molecules. We have also explained what a heteroatom is in line 141.*

10. L34 please state salinity range here or previously

    *Added.*

11. L34 what does inferred biochemical transformations mean? Are these the ones that were measured? It would be more direct to just state which biochemical transformations are being referred to.

*The biochemical transformations are putative gains or losses of molecules based on mass differences among peaks in the spectra. The transformations are not directly measured. They are inferred from the FTICR-MS data by matching the mass differences between pairs of peaks to molecular mass of known biochemical transformations. For example, a mass difference of 99.07 corresponds to gain or loss of the amino acid valine while a difference of 179.06 corresponds to gain or loss of a glucose molecule. We have provided this explanation in the methods (lines 338-343). There are 92 common biochemical transformations (based on commonly observed mass difference associated with biochemical transformations which we evaluated for in our data, and referenced from the following:* (Bailey et al., 2017; Breitling et al., 2006; Graham et al., 2017a, 2018; Stegen et al., 2018). *We have also added a figure (Figure 2)(line 338-340, 1163-1168) to explain our approach to inferring putative biochemical transformations based on mass differences.*

12. L35 which metrics of microbial activity were measured?

*Sorry for any confusion. We did not measure microbial activity, and were careful to point that out (Line 36: "indicate lower microbial activity", and lines 100-103 "These observations suggest that microbial activity usually increases with salinity in soils that were not previously exposed to saline conditions, while simultaneously indicating reduced microbial activity with increasing salinity in soils that have a historical exposure to elevated salinity." We state in lines 597-60 that "decreasing biochemical transformations and heteroatom content (with increasing salinity) imply (but do not quantify) lower microbial activity at higher salinity".*

13. L41 "Null modelling revealed strong influences on dispersal limitation" I am unclear what this means. So the microbial communities were spatially variable or distinct from each other depending on where the samples were taken?

*Strong influences of dispersal limitation influence microbial community composition by restricting the movement of organisms through space that, in turn, allows random demographic events (births and deaths) to cause unstructured divergence in community composition. This unstructured or stochastic divergence is known as ecological drift. In this case, divergence in community composition is not due to deterministic, selective forces systematically causing some taxa to have higher or lower fitness. We have edited the abstract to reflect this with more direct language.*

14. L44 What is a community assembly process? Does this just mean C mineralization, or nitrification, or some other microbially driven process?

*Community assembly processes are those processes that govern the composition of ecological communities (Stegen et al., 2012, 2013, 2015). Assembly processes are either deterministic (selection resulting from different organisms having different levels of fitness*

*for a given set of environmental conditions including abiotic variables and biotic factors related to organismal interactions) or stochastic processes (random birth/death events leading to unstructured divergence in community composition). We have edited the manuscript to be more clearly define these terms and concepts in lines 161-169.*

15. L44 "lack of an association" can the authors be more specific. How were microbial communities measured? PLFA? Molecular techniques? Which part of the microbial communities were compared to C chemistry?

*We have rephrased this sentence to be clearer and more direct. There was no significant relationship (based on regression) between C chemistry and the relative influences of different community assembly processes (as quantified by the bNTI metric, which is detailed in the Methods section). The microbial community composition was determined using 16S rRNA amplicon sequencing, and these data were used to run the null model underlying the bNTI metric (see Methods for our sequencing and null modeling approaches).*

16. L44 "C chemistry" can the authors be more specific? Which C compounds?

*Associations were not evaluated with C compounds. They were evaluated with C chemistry information including Gibbs Free energy, transformation profiles, and heteroatom content, (as written in methods section; lines 316-329). We edited the sentence to give an example of what is meant by C chemistry.*

17. L45 "disconnect btn community and C biogeochem" can you be more specific? What part of the community and biogeochemical processes were disconnected?

*The relative influences of different community assembly process were statistically uncorrelated with C chemistry (i.e., Gibbs Free energy, transformation profiles, and heteroatom content). The Abstract has been edited in line 47-52 for clarity.*

**Introduction**

18. L100 change rates to processes. Rates are not microbially driven, processes are. Which rates/processes are decoupled? Which gas fluxes? CO2 and CH4?

*We have edited the line in the revised manuscript.*

*Gas flux rates are decoupled from dissolved organic carbon concentrations.*

*We are referring to both $CO_2$ and $CH_4$.*
*The sentence has been revised.*

19. L101 Size of C pool… is this referring to the concentration of DOC mentioned in L100? Clarify

*Yes and clarified in line 111-112.*

20. L100-103 How does a decoupling between the C pool size and microbial activity in saline environments suggest it is due to salinity exposure history? Based on how this paragraph is written, it seems like the authors can only say it is due to elevated salinity. Clarify what is meant by salinity exposure history.

*We derive inference from our literature review in the previous paragraph that shows a general trend where soils from historically fresh environments show a positive relationship between $CO_2$ flux and experimentally manipulated salinity. In contrast, soils in historically saline environments show a negative relationship between $CO_2$ flux and experimentally manipulated salinity. This indicates that the history of salinity exposure strongly influences the effect of salinity on $CO_2$ fluxes. We further explain the implications of the salinity exposure history on the resource environment of microbes, bulk C signatures that cannot represent molecular-level changes, and often no observable shifts in bulk C even when a salinity exposure occurs in historically saline or fresh environment. We have rephrased the line to read "Relatively consistent gas flux responses **to changes in salinity….**"Line 109.*

21. L107 Microbial-activity driven??? Needs to be reworded

*Changed to 'microbially driven'.*

22. L98-120 this paragraph starts about discussion between relationships (or lack of) between gas fluxes, DOC, and microbes and ends in a discussion about methods for analyzing chemical constituents of SOC. This should be split up into two paragraphs or reworded to provide better flow. Maybe the first part can be incorporated into the previous paragraph.

*We have edited this paragraph and moved the section to line 180-191.*

23. L135 please define heteroatom as it is not necessarily a common term when describing SOC

*We have defined it in the line (organic compounds containing N, S, P).*

24. L137-138 What is it about increasing salinity that leads to greater heteroatom concentration? This point is unsupported by the first part of the sentence which seems to just be a general statement.

*It is expected that actively growing microbes increase heteroatom containing organic compounds (as indicated by the references cited). Since $CO_2$ fluxes increase with increasing salinity in freshwater systems, we hypothesized that as a historically freshwater system (that began changing to a saltwater system following removal of a culvert in 2014), Beaver Creek soils would also show increasing activity and therefore greater heteroatom content with increasing salinity. We edited this section for clarity, laying out expectations in lines 138-143.*

25. L140 N mining…please be more specific…N uptake from soil? In the form of inorganic or organic N? Is it already available for uptake or do the microbes secrete enzymes to liberate organically bound N in order to take up inorganic N?

*We edited the sentence to reflect a strategy that may require microbes to breakdown organic molecules to extract N (i.e. N mining).*

26. L143 clarify that the flooding that results in marine derived OM is flooding from marine salt water terrestrial systems. I assume the terrestrial ecosystem is freshwater, but up to this point there has been no mention of whether the flooded environment is already saline or is freshwater.

*We revised the line to clarify this as tidal flooding and that the upland site is freshwater.*

27. L150-165 As a reader, I am having trouble following the logic of this paragraph mainly due to the lack of specificity in the use of terms such as community assembly processes, ecological assembly processes, biogeochemical processes, deterministic and stochastic assembly processes, and dispersal processes. Can the authors give examples of what processes they are specifically referring too? It is too general to build a hypothesis off of based on salinity changes in the environment. What is the difference between a community and ecological assembly process? And which can be grouped into deterministic and stochastic categories?

*With respect to community assembly processes we point the reviewer to our responses and revised text discussed above under Reviewer2 comments 13 and 14.*

*We built our hypothesis based on combining two observations from the literature. First, Graham and Stegen (2017) showed that biogeochemical rates are higher when deterministic processes drive community assembly. Second, microbial activity and associated biogeochemical rates have been shown to increase with increasing salinity in historically freshwater systems. Putting these results together provides the hypothesis that the influence of deterministic assembly processes will increase with increasing salinity, due to our study system being historically freshwater (Lines 173-178).*

28. L160 Why subsurface microbial ecology? Are the effects different in soil surface horizons?

*Edited to include both surface and subsurface.*

**Methods**
29. L184 provide lat and long coordinates at the end of the first sentence

*Edited.*

30. L186 Can any information be provided on the extent of inundation onto the landscape? Or the size of the floodplain?

   *The Beaver Creek watershed is 3.8 km². The tidal floodplain makes up 0.5 km² of this total watershed area. Information added in the revised manuscript.*

31. L189 define psu

   *Edited.*

32. L197-199 please provide common names for species as well

   *Agrostis stolinifera (creeping bentgrass), Tsuga heterophylla (Western hemlock), Picea sitchensis (Sitka spruce). Edited.*

33. L204-207 How long were the transects? At what distances along the transects were samples taken?

   *Each transect is roughly 80-90 m. Samples were collected ~35 m apart (information provided in lines 244-245). Edited*

34. L208-209 I prefer to see soil taxonomic information as well as soil series information. It gives readers a choice on what to interpret. I am not that familiar with Ocosta or Mopang soil series so it provides very little information to me about the soil characteristics without having to go look it up on the NRCS.

   *Edited. These are Andisols.*

35. L210 Any idea on water table depth? How deep is the water that pools on the surface?

   *The water table depth in the floodplain is variable both seasonally and throughout tidal cycles/flood events. During floods, there can be almost 1 m of standing water depending on the tide height. During the summer we have observed the water table to be deeper than 60 cm below the ground surface, whereas in the winter it is higher (e.g. 20-30 cm). We have learned from a series of piezometer transects installed across the floodplain that the hydrology of this system is very complex. We are working on a 3-D hydrologic model to describe these dynamics along with salinity, but this effort is beyond the scope of the present manuscript, and unfortunately will not be published soon enough to be referenced here. We will provide the following info for context, while attempting to not lean too heavily on unpublished results in lines 236-239:*

   *"The transects experience periodic inundation episodes which result in surface pooling of tidal water, which can be up to ~1m deep. The water table varies seasonally and during tidal cycles and inundation events, ranging from 0 to ~1m below the ground surface (Ward, unpublished)."*

36. L217-218 It would be nice to know the elevation of the floodplain, inland, and upland transects.

*Elevation provided in edited manuscript (lines 241-243)*

37. L219 Are shallow samples 0-10 cm depth?

*Shallow samples were collected at 10 cm depth.*

38. L224-229 There should be a little bit more detail here on each method, or maybe citations to the methods used at the very least. Provide make, model, company etc. for Lachat. How was pH measured, conductivity, GWC, BD, and porosity?!?! What about pre processing? Was large organic matter removed including roots and litter, or retained. Were samples air dried, sieved, etc..?

*We have added the relevant information in our revised manuscript in lines 256-261. We did not have any litter at the depths we collected the soils from. Sieving ensured removal of roots. All analyses were done on air-dried sieved soils.*

39. L227-229 this doesn't need to be included here. It is in the following sections.

*Edited.*

40. L243 followed by of….check wording

*Edited.*

41. L294-295 It seems like more information should be provided on the microbial DNA procedures.

*All procedures were performed as per* (Bottos et al., 2018) *that has been cited in the manuscript. We provide a brief overview (lines 350-361) of the associated methods and point the reader to Bottos et al. (2018) for additional details.*

**Results**

42. L352 Table S3 is almost unreadable in the small font size

*This is a large file. We can alternately have an excel table uploaded if allowed to do so or have this table be hosted with our data and provide a link. We would appreciate input from the editor on this point.*

43. L392 missing comma after 14%

*Edited.*

44. Why have the authors chosen to not include any taxonomical data on the microbial communities? It seems that this would be very useful information and I assume this information was obtainable from the methods used.

*We agree that it can be useful to know the taxonomic composition but this information is not central to testing our hypotheses. As such, we have included a bar chart of major phyla and provided the OTU table with taxonomic assignments in our data package (See doi in the Data Availability section) so that others can more deeply evaluate taxonomic structure.*

**Discussion**
45. L463-464 Here the authors have at least provided some examples of the biogeochem processes they are interested referring to.

*Please note that we have indicated the same in lines 85-89.*

46. L471 characteristics?

*Edited.*

47. L472 Authors mention spatially structure inputs. I assume this is in reference to land scape variation but it would be helpful to be more specific.

*Yes. We edited this in the revised manuscript in lines 558-560 to read: "Our results are consistent with C chemistry being driven by a combination of spatially-structured inputs driven by landscape structure (i.e., terrestrial inputs further inland, marine inputs further downstream) and…"*

48. L473 What metabolic responses of microbial communities were measured in this study?

*We did not measure microbial metabolic responses. Instead, we find that microbial community composition is not related to SOC thermodynamic properties or indices that reflect microbial activity (i.e., number of biochemical transformations and heteroatom content). As such, any association between microbes and C chemistry must be mediated by changes in microbial metabolism. That is, our data are consistent with the interpretation that it doesn't matter 'who' is there, it matters what metabolisms they are expressing. Additional work is clearly needed that directly examines microbial metabolism, and we revised the manuscript to more directly indicate this need. We added the following text in lines 562-564: "An important caveat is that we did not measure microbial metabolism, but instead infer an influence of microbial metabolism due to microbial composition being independent of C chemistry."*

49. L489 Suffering….awkward wording….Also this appears to be the first mention of forests/tress under stress. Can the authors elaborate on this or provide site level data confirming this?

*Edited. We write about trees under stress in the methods section in lines 218-220 and 583-586. We edited the text for clarity and have provided additional reference from our recent vegetation survey publication at this site.*

50. L494 The authors didn't measure mineral associated C. How then can comments be made about that fraction of the soil C pool? Maybe because these are generally silt and clay rich soils compared to the clearly much more organic surface soils?

    *We edited the methods to indicate that we treated CHCl3-extracted organic carbon as proxy for mineral bound C (line 265) as per (Graham et al., 2017a).*

51. L533 How did the authors determine dispersal limitation? Does this mean that the microbial communities were different between the sites? This would not be surprising but is hard to determine since microbial taxonomic structure was not provided.

    *The influence of dispersal limitation (relative to other community assembly processes) was quantified using a previously established null modeling approach, as discussed in lines 366-388 (see Stegen et al. 2012, 2013, 2015). We edited the associated text for clarity. We chose to focus on ecological community assembly processes rather than look directly at taxonomic composition because of previously published simulation model-based predictions indicating a potential association between assembly processes and biogeochemical function (Graham and Stegen 2017). Null modeling is required because examining taxonomic composition directly does not provide information on community assembly processes.*

52. L542 relatively fast dynamics…..unclear what this means….fast changes in the chemistry of the C? be specific.

    *Edited.*

53. L556-557 I find this statement to be highly speculative given the one sampling date and the lack of measurements of any actual microbial activity metrics. I would argue that there were no measures of biogeochemical functioning in this study, just measures of the outcome of biogeochemical processes (e.g. remaining C compounds, N compounds etc.).

    *We recognize that our study captures one time point and we did not measure microbial activity or biogeochemical function. We are speculative within limits to suggest that community composition in our system is not associated with biogeochemical function. We are citing other papers that show poor association between composition and biogeochemical function. Our sentence structured is purposefully cautions, setup as 'combining our study with these previous…' We added additional language to make it clear that our results are 'consistent with' our inferences and previous literature, though we cannot make unequivocal statements. We also removed 'biogeochemical function' from this sentence. The text in lines 652-654 now reads: "Combining our study with these previous investigations provides evidence that is consistent with (but does not prove) that*

*soil microbial community composition can be independent of C chemistry, though this certainly varies across systems (e.g., Stegen et al. 2018)."*

54. L159 is a more accurate statement…..microbial community (although I think the microbial community structure, abundance of different taxonomic groups, etc. should be shown) was compared to soil C chemistry.

*Line 559 response: we used the bNTI metric to show that microbial community composition and phylogenetic relatedness were not associated with C chemistry. We would also like to guide the reviewer to lines 640-651 that show that community composition may not change while function changes. We now also include a bar plot in in our data package to show the taxonomic groups and provide the OTU table as part of our data package.*

55. L562-563 This is the first time, as far as I can tell, that the authors attempted to define dispersal limitation. This information needs to be given when this is first mentioned in the manuscript.

*We added a brief description in the Introduction with citations to guide readers to resources that explain community assembly processes in detail. In addition, we added explanatory text in the Methods section focused on Ecological Modeling. That text in lines 380-388 reads: "Pairwise community comparisons that do not deviate significantly from the null distribution (i.e., 2> βNTI>-2) indicate the dominance of stochastic processes (including homogenizing dispersal and dispersal limitation), or a scenario in which neither deterministic or stochastic processes dominate (referred to as undominated). Homogenizing dispersal occurs when rate of dispersal between two communities result in community composition becoming relatively similar between the two communities, and potentially overwhelming other assembly processes (e.g., variable selection). Dispersal limitation is the result of very low rates of organismal exchange between communities, which can result in the stochastic divergence of community composition through the accumulated outcomes of random birth/death events (i.e., ecological drift)."*

56. L563-L566 How does restrictive movement of microbial communities in space lead to functional redundancy? It seems like this would actually reduce functional redundancy as spatially restricted microbial communities become more specialized over time especially in salinated and non salinated soils which likely has a marked effect on the microbial community structure.

*Because ecological drift (enabled by dispersal limitation) can lead to the random loss of taxa within local communities, it can result in different communities containing different, but functionally redundant taxa. This has been added to the revised manuscript version in lines 663-665.*

**Tables and Graphs**

57. Figure 1. It would be helpful to have a label for the waterway in the right hand side of the bottom panel. I think that is Beaver Creek but unsure. Maybe this tributary to Johns River does not have a name though?

*Sorry for the confusion—the waterway in the bottom panel is in fact Beaver Creek. The confluence of Beaver Creek and Johns River is not shown in this panel. We have added a label accordingly.*

58. Figure 2 and 3. I recommend color coding the points for each of the three sites so readers can see where they fall out on the regression line.

*Edited. The salinity of the soil samples did not follow a clear spatial gradient.*

59. Table S3 font size should be increased if possible

*This is a large file. We can alternately have an excel table uploaded if allowed to do so or have this table be hosted with our data and provide a link. We would appreciate editorial guidance.*

[revised manuscript text omitted]